# Prominence-Aware Artifact Detection and Dataset for Image Super-Resolution

## Abstract

Generative single-image super-resolution (SISR) is advancing rapidly, yet even state-of-the-art models produce visual artifacts: unnatural patterns and texture distortions that degrade perceived quality. These defects vary widely in perceptual impact—some are barely noticeable, while others are highly disturbing—yet existing detection methods treat them equally. We propose characterizing artifacts by their *prominence* to human observers rather than as uniform binary defects. We present a novel dataset of 1302 artifact examples from 11 SISR methods annotated with crowdsourced prominence scores, and provide prominence annotations for 593 existing artifacts from the DeSRA dataset, revealing that 48% of them go unnoticed by most viewers. Building on this data, we train a lightweight regressor that produces spatial prominence heatmaps. We demonstrate that our method outperforms existing detectors and effectively guides SR model fine-tuning for artifact suppression. Our dataset and code are available at `tinyurl.com/2u9zxtyh`.

## 1. Introduction

Single-image super-resolution (SISR) aims to reconstruct high-resolution (HR) images from low-resolution (LR) inputs and has become a cornerstone of low-level vision tasks. While deep learning and generative adversarial networks (GANs) have greatly improved perceptual quality, they introduce a critical challenge: generation of visually unpleasant artifacts. These artifacts—usually unnatural patterns, smeared faces, and texture distortions—degrade perceived quality and hinder adoption. Even the latest transformer- and diffusion-based methods (Liang et al., 2021; Yu et al., 2024) remain prone to generating artifacts.

Despite SISR's growing popularity, research on detecting

[1]Anonymous Institution, Anonymous City, Anonymous Region, Anonymous Country. Correspondence to: Anonymous Author <anon.email@domain.com>.

Preliminary work. Under review by the International Conference on Machine Learning (ICML). Do not distribute.

SR artifacts remains scarce. LDL (Liang et al., 2022) and DeSRA (Xie et al., 2023) both rely on residual statistics to localize artifacts but differ in supervision: LDL uses HR references and regularizes SR models during training, whereas DeSRA contrasts outputs from the same backbone trained with GAN vs. MSE losses. Approaches such as PAL4VST (Zhang et al., 2023) predict a binary mask from the output image, similarly to semantic segmentation.

These methods rely on manually annotated datasets that contain binary artifact masks. We argue that this limitation is critical: artifacts vary in their prominence to viewers. For example, distortions to regular structures such as buildings, or to recognizable objects such as human faces, easily draw attention and can be distressing to viewers (Figure 1). On the other hand, artifacts on water, grass, and other organic matter can be almost unnoticeable (Figure 2). Treating these different cases as equal carries the risk of overfitting a detection method to less important artifacts while missing the disturbing ones, thus degrading the viewing experience.

To address this limitation, we created a comprehensive dataset of 1302 SR artifact examples generated by 11 contemporary SISR methods from 500 source images, each annotated with a prominence score from extensive crowdsource assessments. We further collected prominence scores for all 593 artifact examples in the DeSRA dataset (originally annotated in lab by Xie et al.) and found that nearly half of these artifacts aren't prominent to most viewers.

Building on our dataset, we propose a prominence-modeling method to detect and quantify SISR artifacts. Our study evaluated existing artifact-detection and image-quality metrics for their ability to predict prominence and trained a lightweight regressor to map their outputs to spatial prominence heatmaps. In parallel, we adapted full-reference methods using the output of RLFN (Kong et al., 2022)—a real-time SR method robust to artifacts—as pseudo ground truth (pseudo-GT), showing that this does not significantly degrade detection quality. This makes our approach, and full-reference baselines, applicable in real-world scenarios where high-resolution ground truth is unavailable.

Our main contributions are the following:

1. We present the first dataset of its kind, containing 1302 SISR artifact examples with crowdsourced promi-

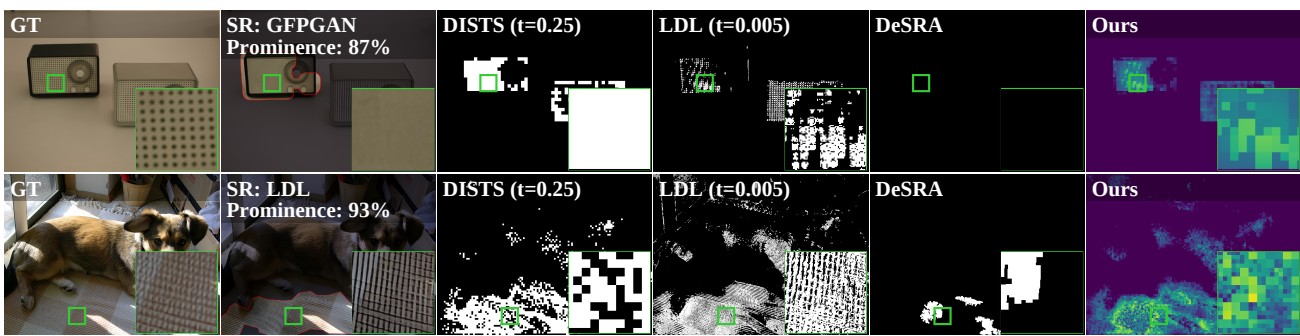

*Figure 1.* Examples of prominent artifacts detected by the proposed method. Top: example from our proposed dataset. GFPGAN failed to restore holes on the radio panel. Bottom: example from the DeSRA dataset. LDL produced an irregular line pattern on the carpet. Prominence denotes the percentage of annotators confirming the artifact in the highlighted area (Section 3).

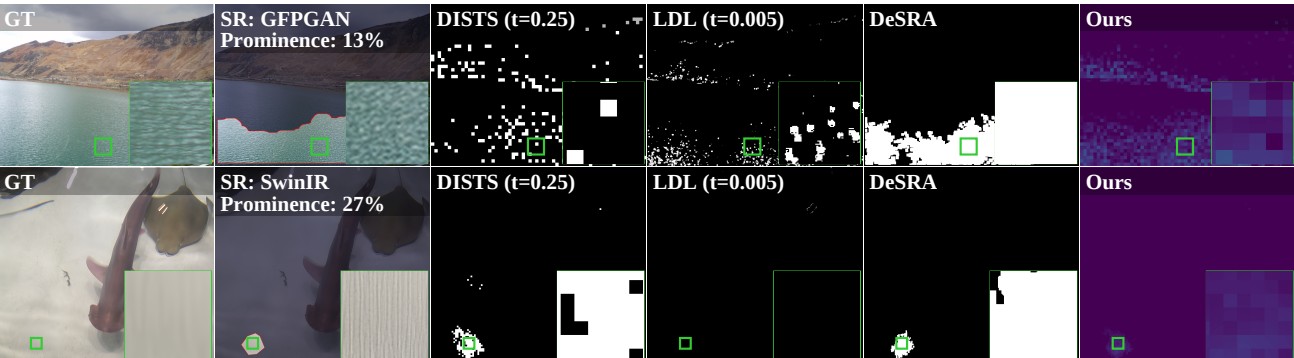

*Figure 2.* Non-prominent artifacts detected by existing methods. Top: example from our proposed dataset. The water surface was incorrectly restored by GFPGAN producing an artifact, but it's a natural surface, so this artifact is not prominent to humans. Bottom: example from the DeSRA dataset. SwinIR produced a texture artifact (vertical lines) on the floor, but it's in a non-salient region, so it's not prominent to humans. Our method correctly marks both examples with low prominence.

nence annotations; it features diverse distortions from 11 SISR methods. To adapt for visual inspection the tight masks that artifact-detection methods produce, we propose a simple mask-postprocessing algorithm. We additionally collected prominence annotations for the 593 artifact examples in the DeSRA dataset (Xie et al., 2023).

2. We propose a perceptual-prominence modeling method that detects and quantifies artifacts in SR images and that outperforms existing approaches, judging by thorough objective and subjective evaluation. We show that our method, relative to prior art, is more applicable to fine-tuning of SR models for artifact reduction. Furthermore, we demonstrate that real-time SR can be a useful pseudo-GT for full-reference metrics.

3. Using our artifact-detection method and existing alternatives, along with our prominence annotation methodology, we analyzed 11 SR methods for their propensity to generate artifacts. We show that even the newest high-quality methods such as SUPIR (Yu et al., 2024) are very susceptible to this problem.

## 2. Related work

**Single-image super-resolution (SISR)** has undergone extensive study over the past decade. Early methods optimized pixel-wise losses such as $L_1$ and $L_2$, which improved PSNR/SSIM but failed to recover realistic details. GAN-based approaches improved perceptual sharpness and introduced realistic degradation pipelines for training (Ledig et al., 2017; Zhang et al., 2021; Wang et al., 2021b).

Recent SISR developments span Transformer architectures, diffusion models, and hybrid generative priors. Transformers like HAT (Chen et al., 2023) and DRCT (Hsu et al., 2024) use multiscale attention and dense residual connections for efficient quality enhancement. Diffusion methods such as ResShift (Yue et al., 2023) and SinSR (Wang et al., 2024b) accelerate sampling via residual shifting and deterministic distillation. StableSR (Wang et al., 2024a) leverages Stable Diffusion (Rombach et al., 2022) priors for upscaling, and SUPIR (Yu et al., 2024) scales models and data for text-guided photo-realistic restoration. These advances, however, introduce the challenge of visual artifacts.

**SISR evaluation** has traditionally relied on full-reference

metrics such as PSNR and SSIM (Wang et al., 2004), which assess reconstruction fidelity but correlate poorly with perceptual quality—especially for GAN-based outputs where details and artifacts are entangled. No-reference and perceptual metrics such as NRQM (Ma et al., 2017), LPIPS (Zhang et al., 2018), and DISTS (Ding et al., 2022) better align with human perception and are now widely adopted in SR benchmarks. Some techniques aim to make metrics more artifact-resistant: ERQA (Kirillova et al., 2022) evaluates detail restoration by matching edges in reference and test images. However, in practice existing metrics still fall well short of matching human perception (Borisov et al., 2025).

**Detection and mitigation of SISR artifacts** has garnered increasing attention because these artifacts reduce perceptual quality. LDL (Liang et al., 2022) predicts pixel-level artifact maps from local residual statistics. Xie et al. (2023) introduced an in-lab annotated dataset with binary SR artifact masks and, building on it, proposed DeSRA that contrasts GAN-SR and MSE-SR outputs to identify artifact-prone regions, then fine-tunes the SR model on a few samples to suppress those regions.

A complementary line of work treats artifact detection as segmentation, training networks on datasets with pixel-level defect maps. Given only an input image, these models predict an artifact mask. Approaches such as PAL4Inpainting (Zhang et al., 2022) and PAL4VST (Zhang et al., 2023) show strong generalization across generative vision tasks by localizing perceptual artifacts.

Concurrently, Ren et al. (2025) propose Hallucination Score that uses a multimodal LLM to provide an image-level hallucination rating for SR outputs, showing strong alignment with human judgments. The main drawback of this approach is that it lacks spatial localization, which is critical for downstream tasks such as artifact mitigation, SR model fine-tuning, and for handling cases where different regions of an image exhibit different types of artifacts.

Despite these advances, most approaches remain limited by the shortage of annotated datasets that explicitly focus on SR artifacts. While datasets such as DeSRA provide binary artifact masks, most methods remain constrained by the absence of richer annotations (e.g., prominence levels), which limits their generalization and robustness in real-world scenarios. Our work addresses this shortfall by introducing a novel dataset annotated with artifact regions and prominence scores, along with a prominence-aware detection method that supports SR fine-tuning and reveals that even the latest models like SUPIR remain highly artifact-prone.

## 3. Artifact dataset

Existing datasets such as DeSRA (Xie et al., 2023) contain only binary artifact masks, without information on how

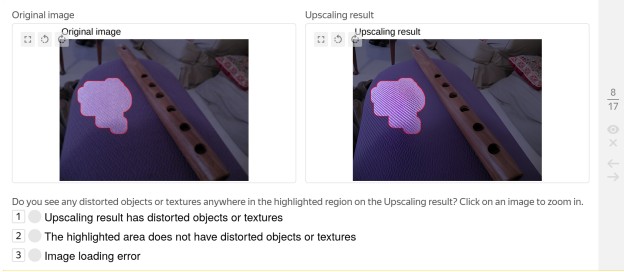

*Figure 3.* Viewer interface for subjective data collection.

noticeable the artifacts are to viewers. To enable research on artifact *prominence*, we introduce a dataset of 1302 artifact examples, each annotated with both a binary mask and a prominence score derived from crowdsourced assessments.

Our dataset is based on Open Images (Kuznetsova et al., 2020), a diverse collection of about 300,000 natural images (CC BY 2.0). We randomly selected 2101 source photos, each 768×1024 pixels. These photos then underwent 4× bicubic downsampling followed by upsampling with 11 popular SR methods (Yu et al., 2024; Wang et al., 2021b; Chen et al., 2023; Hsu et al., 2024; Yue et al., 2023; Wang et al., 2024b;a; Cai et al., 2019; Liang et al., 2021; Wang et al., 2021a; Wu et al., 2024), yielding 23,111 images for artifact search. We obtained the initial binary artifact masks through manual annotation, and by running existing visual-quality metrics (SSIM, DISTS, LPIPS) and artifact-detection algorithms (LDL, DeSRA, in-progress versions of our method). For each of these algorithms we selected the 100 strongest detected distortions and manually discarded images without artifacts. The remaining images underwent crowdsourced prominence annotation, resulting in 697 artifact examples. The evaluation in Section 5.4 yielded 605 more examples.

We additionally collected prominence annotations for all 593 images from the DeSRA artifact dataset.

### 3.1. Crowdsourced annotation setup

We used Toloka.ai to crowdsource the data collection. Participants view pairs of images labeled "Original" and "Upscaled," with the artifact region visually highlighted. We ask them whether the highlighted region contains a distorted object or texture. Figure 3 shows an example question.

Every image receives a ranking by 30 different participants. We compute prominence as the proportion of votes indicating the artifact is present. Before receiving access to the main questions, participants must answer four training questions, for which the correct answers are explained, followed by four test questions with hidden correct answer. Afterward, to ensure integrity, every group of 20 questions contains 4 random control questions. All responses from participants who mistakenly answer any control question

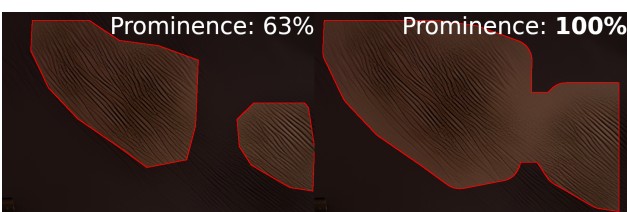

*Figure 4.* Example of our postprocessing technique increasing artifact visibility. On the left is the original mask from the DeSRA dataset; on the right is the mask after postprocessing.

*Table 1.* Comparison between the DeSRA and our artifact dataset.

|  | DeSRA | Ours |
| --- | --- | --- |
| # SR methods | 3 | **11** |
| # Artifacts | 593 | **1302** |
| Mask source | in-lab | automatic |
| Label type | binary | **prominence** |

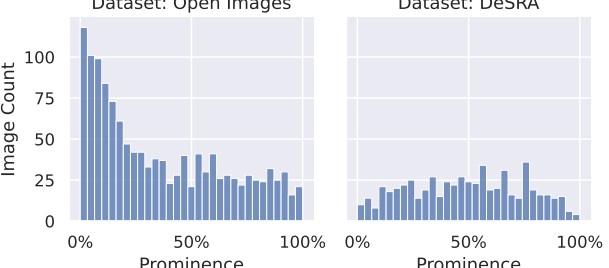

*Figure 5.* Prominence distribution for our dataset (based on Open Images) and for the DeSRA dataset (for which we collected prominence annotations).

are discarded. In total, 596 participants successfully completed the annotation. Section B analyses the impact of the participant count on the answer variability.

### 3.2. Mask postprocessing

An artifact-detection method aims to output a tight mask around an artifact, since doing so is more useful for further analysis and for downstream tasks such as automatic correction. But tight masks make it harder to visually judge whether the masked area contains an artifact. To remedy this, we apply morphological operations to the masks before showing them to participants:

1. Open with a 25×25 square kernel to remove small dots in the mask.
2. Dilate with a 64×64 circular kernel so the mask includes context around an artifact.
3. Close with a 25×25 square kernel to eliminate holes and step away from the image borders.

The example in Figure 4 shows how a tight mask makes an artifact harder to notice compared with our postprocessing result. We verified the validity of this step by running the crowdsourced prominence annotation twice on the DeSRA dataset: once with our postprocessing and once with unmodified masks. For this comparison, separate groups of participants conducted the annotations, with matching question order. As a result, the mean artifact prominence for the entire DeSRA dataset was 49.4% with our postprocessing and 47.7% with the original masks, indicating the postprocessing helps viewers judge an artifact's presence.

### 3.3. Dataset analysis

Table 1 and Figure 5 compare our dataset to DeSRA and show prominence distributions.

Our dataset has a bias toward zero prominence (no or unnoticeable artifacts); the reason is that we seeded the masks with results from existing image-quality metrics poorly adapted to finding artifacts and from existing artifact-detection methods that lack the ability to differentiate between barely visible and highly visible artifacts. This bias has implications for objective evaluation, potentially

skewing results, and can induce overfitting during model training—a concern we address in Section I.4.

The DeSRA dataset shows a more balanced distribution, centered around moderate artifact prominence. Notably, despite its lab annotations, almost half of them have a prominence below 50%—that is, most viewers fail to notice them. This confirms that binary masks are insufficient for accurate SR-artifact evaluation.

## 4. Artifact prominence metric

Our goal is predicting a spatial artifact prominence heatmap for a super-resolved image from its low-resolution input, where higher values indicate more severe artifacts (more prominent to human observers). Figure 6 outlines our method, which aggregates three features from existing quality and artifact-detection metrics via a lightweight fusion module. The resulting model finds prominent artifacts more efficiently than any individual feature or other approaches.

### 4.1. Input features

We selected features based on their proven performance for evaluating and detecting texture distortions. These features estimate not only the visual quality, but also the structural similarity between the reference and the upscaled image.

The first feature is DISTS (Ding et al., 2022), a visual-image-quality metric which accounts for texture distortions and their perceptual impact. As DISTS is trained on natural images, it effectively detects unnatural degradations like SR artifacts. DISTS produces a single image-level score,

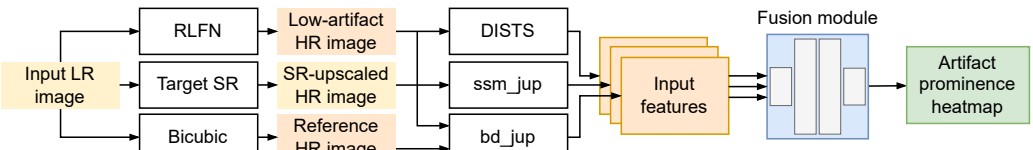

*Figure 6.* Architecture of the proposed artifact prominence metric. The input image is upscaled by the target SR and by RLFN as described in Section 4.3. Then, we compute three features described in Section 4.1. Finally, we run our fusion module described in Section 4.2.

so we computed it block-wise in 16×16-pixel blocks, the minimum input size of the metric.

The second feature, which we call *ssm_jup*, is adapted from the small-color-artifact detector from Tsereh et al. (2024), itself based on LDL (Liang et al., 2022). It targets small-scale distortions and was shown to be effective for finding JPEG AI artifacts. To capture texture distortions, we modify the detector to use all RGB channels rather than only chromatic U and V components. Like LDL, this feature requires a reference image upscaled by an artifact-resistant method; we chose bicubic interpolation for this input.

The last feature, *bd_jup*, is a weighted sum of LPIPS (Zhang et al., 2018) and ERQA (Kirillova et al., 2022) applied block-wise. LPIPS measures how well the upscaled image preserves perceptual quality, and is widely used in SR evaluation. Meanwhile, ERQA assesses the preservation of object details and boundaries. For LPIPS, we used 32×32-pixel blocks with stride of 16. ERQA uses 8×8 blocks with no overlap. LPIPS is weighted 3:2 compared with ERQA.

Section A details the implementation of *ssm_jup* and *bd_jup*.

### 4.2. Fusion module

We experimented with several architectures for fusing the features described above into a single prominence prediction, including CNN-based and tree-based models (see Sections I.6 and I.7). A shallow multilayer perceptron (MLP) achieved the best overall performance, so we adopted it as our feature fusion module.

The MLP takes as input the feature values, passes them through three fully connected layers (3-128-128-1) with ReLU activations, and outputs a single prominence value. It independently processes each pixel of the input-feature heatmaps, yet still captures broader context since the features themselves encode both the pixel's neighborhood and wider image-level information.

### 4.3. Adapting full-resolution metrics with real-time-SR pseudo-GT

Full-reference metrics provide more-accurate detail restoration quality scores for SR by using pixel-level information from the reference image. The use of such metrics in SR creates difficulties, however, since the SR-output resolution

is higher than that of the original low-resolution frame.

To employ full-reference metrics, we propose the following pipeline. We applied a lightweight SR method to the original low-resolution frame, thereby obtaining a pseudo-GT, and then calculated the metric for this pseudo-GT and the SR output. We noticed that real-time SR methods, such as SPAN (Wan et al., 2024) and RLFN (Kong et al., 2022), produce outputs that, despite trailing heavier SR models in visual quality, are devoid of major visual artifacts. When serving as pseudo-GT for full-reference metrics, the artifact-detection performance drop is small compared with using the original HR frames, as Section 5.2 shows. This approach enables our method to serve in real-world upscaling where the high-resolution GT frames are unavailable.

### 4.4. Training

We train our fusion module using Adam on a training subset of 374 artifact examples from our dataset described in Section 3. The model predicts a prominence value for each pixel of the input image. We compute the mean predicted prominence inside and outside the binary artifact mask from the dataset. The training loss consists of two $L_2$ components:

$$\mathcal{L} = L_2(\text{MeanInside}, \text{GT Prominence}) \\ + L_2(\text{MeanOutside}, 0). \tag{1}$$

The model is trained to predict the ground-truth prominence value inside the binary mask, and 0 (no artifact) outside it. Thanks to small model size, the training converges quickly, usually in around 10–15 epochs. One training epoch takes about 13 seconds on an Nvidia RTX 3090 GPU.

## 5. Experiments

Our evaluation comprises multiple steps. Section 5.1 provides an overview of our approach to objectively evaluating artifact-detection methods using our prominence dataset, then Section 5.2 compares our method to existing work with objective scores. Section 5.3 validates our method's generalization across SR architectures. Next, Section 5.4 evaluates methods on the primary downstream task: finding artifacts prominent to human viewers. Finally, Section 5.5 evaluates methods on the secondary downstream task: reducing SR proneness to artifacts.

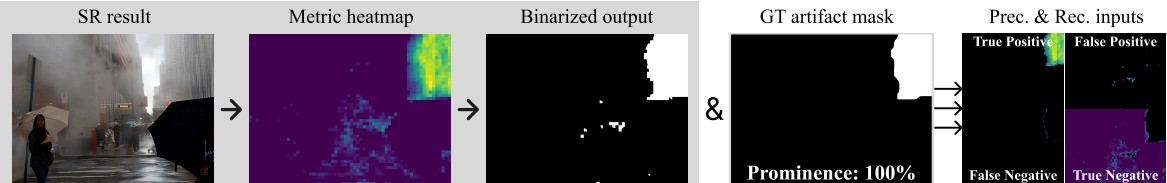

*Figure 7.* Overview of our objective evaluation pipeline. Metric heatmaps are binarized and compared with GT artifact masks to obtain TP/FP/FN/TN regions. Raw heatmap values within these regions are then used to compute soft precision and recall, taking into account crowdsourced artifact prominence.

### 5.1. Objective evaluation methodology

Our evaluation follows the standard binary classification methodology—binarizing detection outputs with thresholds (selected for each method in evaluation) and computing TP, FP, FN—but modifies the precision and recall to weight detections by their prominence scores (Figure 7). Following (Rachakonda & Bhatnagar, 2021; Harju & Mesaros, 2023), we treat artifact labels as graded values rather than binary. In our setting, each label is a prominence score rather than a probability of class membership. A low prominence score indicates that the masked area contains no artifact, so a positive detection in such cases should be penalized. We implement this using a margin $\kappa = 0.3$, yielding:

$$Prec^{pr} = \frac{\sum_{i \in \text{images}} TP_i * (\mathrm{p}_i - \kappa)}{\sum_{i \in \text{images}} (TP_i + FP_i)};$$
$$Rec^{pr} = \frac{\sum_{i \in \text{images}} TP_i * (\mathrm{p}_i - \kappa)}{\sum_{i \in \text{images}} (TP_i + FN_i)}. \qquad (2)$$

We set $\kappa = 0.3$ to reward detection of less-prominent artifacts: if more than 30% of viewers reported the artifact, we consider it prominent enough to detect. This choice was guided by dataset statistics and observed viewer consistency.

From $Prec^{pr}$ and $Rec^{pr}$, we compute F1-score and PR-AUC. Unlike standard binary classification, these scores may fall outside $[0, 1]$ (see Section G). However, they remain suitable for ranking methods relative to each other.

### 5.2. Objective prominence-metric evaluation

We evaluate our method using both our proposed methodology, and the methodology from DeSRA (Xie et al., 2023), to avoid bias towards our own dataset and scoring. Following DeSRA, we selected binarization thresholds for all methods by maximizing the Precision × Recall product on the prominent subset of the DeSRA dataset. The thresholds are shown in all tables as `t=0.xx`.

Table 2 compares our method with other approaches on the basis of our prominence datasets, under different choices of reference input as described in Section 4.3. These include original high-resolution frames, the lightweight SR models SPAN (Wan et al., 2024) and RLFN (Kong et al., 2022), and,

for DeSRA where HR frames are unavailable, the authors' MSE-SR model trained with an MSE loss. Since the MSE-SR weights were not released, this reference cannot be used in other experiments.

Our method delivers the most consistent performance across all experiment settings, as reflected by its best average ranking in PR-AUC. These results also confirm that pseudo-GT remains practical for artifact detection. We adopt RLFN pseudo-GT for all other experiments in the paper.

Next, we followed the DeSRA comparison methodology and computed binary precision, recall, and intersection-over-union (IoU) on the same datasets (no $\kappa$ margin). This comparison, however, only considered a subset with prominence values above 50%; doing so yields a more accurate evaluation, as artifacts with low prominence values are barely noticeable and should be considered false positives. Table 3 shows the results. Our method outperforms competitors in most experiment settings.

We also evaluated our method on a learning-based compression task, as described in Section H. You can find $Prec^{pr}$ and $Rec^{pr}$ values for all methods in Section I.8.

### 5.3. Evaluating generalization across SR architectures

To evaluate our method's ability to generalize across different SR models, we conduct a cross-validation experiment where we train on artifact examples from two SR families and validate on a held-out third family. We partition our artifact dataset by the SR family (CNN, transformer, or diffusion), train three separate models each excluding one family, and evaluate them on the excluded family's artifacts.

Table 4 shows the results, including existing methods for comparison. Our models show the most consistent performance demonstrating that the learned artifact representations transfer across SR architectures. This suggests that artifacts from different SR families share common perceptual characteristics that our method successfully captures.

### 5.4. Evaluating robustness to SR artifacts

We evaluated the robustness of 11 popular SR models to generating artifacts, following the preparation process from Sec-

*Table 2.* Results on the full proposed and DeSRA datasets. PR-AUC aggregates over thresholds so it is shown once. The last column shows the method's average ranking by PR-AUC compared to others.

| | Proposed Dataset | | | | | | DeSRA Dataset | | | | | | Avg. |
| | Original HR | | SPAN | | RLFN | | MSE-SR | | SPAN | | RLFN | | Rank$^{\downarrow}$ |
| Reference Input Method | F1-score$^{\uparrow}$ | PR-AUC$^{\uparrow}$ | F1-score$^{\uparrow}$ | PR-AUC$^{\uparrow}$ | F1-score$^{\uparrow}$ | PR-AUC$^{\uparrow}$ | F1-score$^{\uparrow}$ | PR-AUC$^{\uparrow}$ | F1-score$^{\uparrow}$ | PR-AUC$^{\uparrow}$ | F1-score$^{\uparrow}$ | PR-AUC$^{\uparrow}$ | PR-AUC |
|---|---|---|---|---|---|---|---|---|---|---|---|---|---|
| LDL (t=0.005) | 0.0275 | 0.0039 | 0.0197 | 0.0034 | 0.0035 | 0.0008 | 0.1670 | 0.0518 | **0.1618** | **0.0486** | 0.1622 | 0.0503 | 4.7 |
| SSIM (t=0.55) | 0.0140 | 0.0022 | 0.0359 | 0.0083 | 0.0344 | 0.0054 | 0.1828 | 0.0548 | 0.1488 | 0.0372 | 0.1786 | 0.0551 | 3.7 |
| LPIPS (t=0.25) | 0.0352 | 0.0049 | 0.0418 | 0.0042 | 0.0364 | 0.0043 | 0.1392 | 0.0349 | 0.1371 | 0.0300 | 0.1462 | 0.0389 | 5.2 |
| ERQA (t=0.55) | 0.0028 | 0.0001 | -0.0137 | -0.0002 | -0.0195 | -0.0004 | 0.0396 | 0.0026 | 0.0474 | 0.0017 | 0.0399 | 0.0025 | 9.8 |
| PAL4Inpaint (bin., no-ref) | 0.0117 | N/A | 0.0117 | N/A | 0.0117 | N/A | 0.0609 | N/A | 0.0609 | N/A | 0.0609 | N/A | N/A |
| PAL4VST (bin., no-ref) | 0.0062 | N/A | 0.0062 | N/A | 0.0062 | N/A | 0.0054 | N/A | 0.0054 | N/A | 0.0054 | N/A | N/A |
| PaQ-2-PiQ (t=65, no-ref) | 0.0073 | 0.0000 | 0.0073 | 0.0000 | 0.0073 | 0.0000 | 0.0156 | 0.0006 | 0.0156 | 0.0006 | 0.0156 | 0.0006 | 10.2 |
| TOPIQ (t=0.5, no-ref) | 0.0071 | 0.0000 | 0.0071 | 0.0000 | 0.0071 | 0.0000 | 0.0160 | 0.0025 | 0.0160 | 0.0025 | 0.0160 | 0.0025 | 10.0 |
| DISTS (t=0.25) | 0.0555 | 0.0062 | **0.0706** | 0.0085 | **0.0706** | 0.0082 | 0.1628 | 0.0376 | 0.1071 | 0.0213 | 0.1637 | 0.0457 | 4.2 |
| bd_jup (t=0.1) | 0.0043 | 0.0027 | 0.0105 | 0.0013 | 0.0074 | 0.0017 | 0.1175 | 0.0230 | 0.0920 | 0.0135 | 0.1181 | 0.0244 | 7.2 |
| ssm_jup (t=0.2) | 0.0251 | 0.0012 | 0.0144 | 0.0011 | 0.0180 | 0.0009 | 0.1769 | 0.0377 | 0.1426 | 0.0251 | 0.1690 | 0.0346 | 6.8 |
| DeSRA (t=0.3) | 0.0405 | 0.0068 | 0.0371 | **0.0154** | 0.0315 | **0.0120** | 0.1752 | 0.0579 | 0.1274 | 0.0273 | 0.1696 | 0.0550 | 2.3 |
| **Ours (t=0.15)** | 0.0355 | **0.0121** | 0.0325 | 0.0075 | 0.0312 | 0.0075 | 0.1780 | **0.0617** | 0.1235 | 0.0398 | 0.1737 | **0.0605** | 2.0 |
| **Ours (t=0.3)** | **0.0559** | | 0.0310 | | 0.0334 | | **0.1907** | | 0.1540 | | **0.1902** | | **2.0** |

*Table 3.* Results on the prominent subset of the proposed and DeSRA datasets.

| | Proposed Dataset | | | | | | DeSRA Dataset | | | | | | Avg. |
| | Original HR | | SPAN | | RLFN | | MSE-SR | | SPAN | | RLFN | | Rank$^{\downarrow}$ |
| Reference Input Method | IoU$^{\uparrow}$ | PR-AUC$^{\uparrow}$ | IoU$^{\uparrow}$ | PR-AUC$^{\uparrow}$ | IoU$^{\uparrow}$ | PR-AUC$^{\uparrow}$ | IoU$^{\uparrow}$ | PR-AUC$^{\uparrow}$ | IoU$^{\uparrow}$ | PR-AUC$^{\uparrow}$ | IoU$^{\uparrow}$ | PR-AUC$^{\uparrow}$ | PR-AUC |
|---|---|---|---|---|---|---|---|---|---|---|---|---|---|
| LDL (t=0.005) | 0.1043 | 0.1387 | 0.1788 | 0.3110 | 0.1779 | 0.3361 | 0.3724 | 0.4687 | 0.3896 | 0.3418 | 0.3443 | 0.4182 | 5.0 |
| SSIM (t=0.55) | 0.3460 | 0.3803 | 0.2642 | 0.3862 | 0.2437 | 0.3710 | 0.5327 | 0.5730 | **0.4590** | 0.3861 | 0.5243 | 0.6051 | 2.3 |
| LPIPS (t=0.25) | 0.2621 | 0.3861 | 0.1340 | 0.3044 | 0.1324 | 0.2971 | 0.3759 | 0.4488 | 0.3450 | 0.3193 | 0.4094 | 0.5014 | 4.7 |
| ERQA (t=0.55) | 0.2495 | 0.1220 | 0.1670 | 0.1052 | 0.1492 | 0.1052 | 0.0523 | 0.0100 | 0.0684 | 0.0154 | 0.0514 | 0.0087 | 10.0 |
| PAL4Inpaint (bin., no-ref) | 0.0753 | N/A | 0.0753 | N/A | 0.0753 | N/A | 0.1139 | N/A | 0.1139 | N/A | 0.1139 | N/A | N/A |
| PAL4VST (bin., no-ref) | 0.0463 | N/A | 0.0463 | N/A | 0.0463 | N/A | 0.0140 | N/A | 0.0140 | N/A | 0.0140 | N/A | N/A |
| PaQ-2-PiQ (t=65, no-ref) | 0.0782 | 0.0344 | 0.0782 | 0.0344 | 0.0782 | 0.0344 | 0.0305 | 0.0285 | 0.0305 | 0.0285 | 0.0305 | 0.0285 | 10.5 |
| TOPIQ (t=0.5, no-ref) | 0.1406 | 0.0860 | 0.1406 | 0.0860 | 0.1406 | 0.0860 | 0.0424 | 0.1069 | 0.0424 | 0.1069 | 0.0424 | 0.1069 | 9.5 |
| DISTS (t=0.25) | 0.3525 | 0.2619 | 0.2820 | 0.3242 | 0.2783 | 0.3386 | 0.4919 | 0.4408 | 0.3479 | 0.2290 | 0.5016 | 0.5428 | 5.0 |
| bd_jup (t=0.1) | 0.2843 | 0.3311 | 0.2475 | 0.2342 | 0.2434 | 0.2275 | 0.3580 | 0.1609 | 0.2798 | 0.1221 | 0.3625 | 0.1773 | 7.2 |
| ssm_jup (t=0.2) | 0.2368 | 0.2127 | 0.2133 | 0.2273 | 0.2221 | 0.2411 | 0.4032 | 0.3889 | 0.3770 | 0.2737 | 0.3930 | 0.3646 | 7.0 |
| DeSRA (t=0.3) | 0.2560 | 0.3173 | 0.1296 | 0.3358 | 0.1205 | 0.3025 | 0.5277 | **0.6928** | 0.3707 | 0.2910 | 0.5082 | **0.6614** | 3.3 |
| **Ours (t=0.15)** | 0.3639 | **0.4756** | **0.3018** | **0.3931** | **0.2903** | **0.3829** | **0.5420** | 0.6104 | 0.4010 | **0.3874** | **0.5301** | 0.6031 | **1.5** |
| **Ours (t=0.3)** | **0.3669** | | 0.2357 | | 0.2311 | | 0.4866 | | 0.4374 | | 0.5049 | | |

*Table 4.* Cross-validation results on the prominent subset.

| Val. Family | CNN | | Transformer | | Diffusion | |
| Method | IoU$^{\uparrow}$ | PR-AUC$^{\uparrow}$ | IoU$^{\uparrow}$ | PR-AUC$^{\uparrow}$ | IoU$^{\uparrow}$ | PR-AUC$^{\uparrow}$ |
|---|---|---|---|---|---|---|
| LDL (t=0.005) | 0.100 | 0.258 | 0.027 | 0.154 | 0.133 | 0.256 |
| SSIM (t=0.55) | 0.218 | 0.416 | **0.236** | 0.337 | 0.222 | **0.530** |
| LPIPS (t=0.25) | 0.138 | 0.386 | 0.054 | 0.334 | 0.150 | 0.496 |
| ERQA (t=0.55) | 0.111 | 0.131 | 0.081 | 0.112 | 0.114 | 0.167 |
| PaQ-2-PiQ (t=65) | 0.031 | 0.027 | 0.023 | 0.037 | 0.023 | 0.031 |
| TOPIQ (t=0.5) | 0.173 | 0.106 | 0.143 | 0.088 | 0.052 | 0.100 |
| DISTS (t=0.25) | 0.187 | 0.292 | 0.165 | 0.299 | 0.291 | 0.416 |
| bd_jup (t=0.1) | 0.129 | 0.318 | 0.168 | 0.242 | 0.177 | 0.344 |
| ssm_jup (t=0.2) | 0.158 | 0.278 | 0.103 | 0.174 | 0.280 | 0.309 |
| DeSRA (t=0.3) | 0.158 | 0.372 | 0.063 | 0.224 | 0.170 | 0.424 |
| **Ours T+D (t=0.25)** | **0.239** | **0.470** | — | — | — | — |
| **Ours C+D (t=0)** | — | — | 0.231 | **0.419** | — | — |
| **Ours C+T (t=0.5)** | — | — | — | — | **0.331** | 0.482 |

*Table 5.* Crowd-sourced prominence results across SR models.

| SR | Family | Masks Found | Mean Prominence$^{\downarrow}$ | Conf. Masks Found$^{\downarrow}$ |
|---|---|---|---|---|
| DRCT (Hsu et al., 2024) | Transformer | 43 | **11.85%** | 1 |
| HAT-L (Chen et al., 2023) | Transformer | 53 | 13.53% | 1 |
| SinSR (Wang et al., 2024b) | Diffusion | 60 | 19.39% | 3 |
| ResShift (Yue et al., 2023) | Diffusion | 60 | 20.22% | 7 |
| StableSR (Wang et al., 2024a) | Diffusion | 60 | 28.55% | 14 |
| RealSR (Cai et al., 2019) | CNN | 51 | 28.70% | 10 |
| SeeSR (Wu et al., 2024) | Diffusion | 61 | 32.34% | 14 |
| GFPGAN (Wang et al., 2021a) | CNN | 45 | 32.74% | 11 |
| SwinIR (Liang et al., 2021) | Transformer | 59 | 41.09% | 17 |
| SUPIR (Yu et al., 2024) | Diffusion | 70 | 45.29% | 20 |
| RealESRGAN (Wang et al., 2021b) | CNN | 61 | 48.42% | 19 |

tion 3, but with no manual mask curation. For each metric, we selected the 10 strongest artifacts per SR, yielding 653 in total. We then collected prominence values for these masks via crowdsourcing. Our results report the total number of artifact masks that the metrics produced, their mean prominence, and the number of "confident" masks corresponding to highly visible artifacts (50% prominence or higher).

Table 5 shows the results grouped by SR model. DRCT (Hsu et al., 2024) and HAT-L (Chen et al., 2023), both Transformer based, show excellent results with nearly zero promi-

nent artifacts detected. Next are three diffusion-based methods: SinSR (Wang et al., 2024b), ResShift (Yue et al., 2023), and StableSR (Wang et al., 2024a). Then, Table 7 shows the results grouped by artifact-detection method. Ours shares first place with DISTS in number of confident masks found while beating it in mean artifact prominence. DeSRA ranked third in this evaluation. Section F provides extra crowd-sourced evaluation on other SR datasets.

Note that LDL with a 0.005 threshold finds highly visible artifacts, but it trails far behind other methods in total number of confident masks. To account for both of these scores, we multiplied them analogously to the precision × recall prod-

*Table 6.* Results of fine-tuning SR models on artificial GT constructed with different methods.

| Target SR | LDL (Liang et al., 2022) | | | RealESRGAN (Wang et al., 2021b) | | | SwinIR (Liang et al., 2021) | | | Avg. |
| --- | --- | --- | --- | --- | --- | --- | --- | --- | --- | --- |
| **Method** | $\Delta$IoU$^\uparrow$ | Add$_{img}^\downarrow$ | Rem$_{img}^\uparrow$ | $\Delta$IoU$^\uparrow$ | Add$_{img}^\downarrow$ | Rem$_{img}^\uparrow$ | $\Delta$IoU$^\uparrow$ | Add$_{img}^\downarrow$ | Rem$_{img}^\uparrow$ | **Rank**$^\downarrow$ |
| DISTS (t=0.25) | 27.00 | 33.51 | 11.70 | 33.47 | 32.66 | 15.58 | **15.23** | 26.47 | 13.73 | 3.3 |
| LDL (t=0.005) | 17.26 | 90.43 | 6.91 | 18.32 | 88.94 | 11.56 | 4.22 | 77.94 | 21.08 | 4.9 |
| LPIPS (t=0.25) | 25.18 | **15.43** | 19.68 | 33.66 | **8.54** | 34.17 | 11.21 | **2.94** | 53.43 | 2.2 |
| ERQA (t=0.55) | 0.67 | 100.00 | 0.00 | 1.22 | 99.50 | 0.00 | 0.19 | 98.04 | 0.00 | 6.0 |
| DeSRA | 29.18 | 54.26 | 25.00 | 33.66 | 34.17 | **56.78** | 8.98 | 32.84 | 28.43 | 2.9 |
| **Ours** (t=0.3) | **34.71** | 20.74 | **45.21** | **38.01** | 14.57 | 55.78 | 11.86 | 6.37 | **57.35** | **1.6** |

*Table 7.* Crowd-sourced prominence results across artifact detection methods.

| Method | Masks Found | Mean Prominence$^\uparrow$ | Conf. Masks Found$^\uparrow$ | Comb. Score$^\uparrow$ |
| --- | --- | --- | --- | --- |
| LDL (t=0.005) | 12 | 77.11% | 11 | 8.48% |
| bd_jup (t=0.1) | 110 | 17.03% | 13 | 2.21% |
| LDL (t=0.0005) | 51 | 34.84% | 15 | 5.23% |
| LDL (t=0.001) | 40 | 43.00% | 16 | 6.88% |
| ssm_jup (t=0.15) | 110 | 23.09% | 20 | 4.62% |
| SSIM (t=0.55) | 74 | 36.62% | 26 | 9.52% |
| DeSRA | 110 | 32.03% | 31 | 9.93% |
| DISTS (t=0.25) | 108 | 38.80% | **38** | 14.75% |
| **Ours** (t=0.3) | 99 | 41.25% | **38** | **15.67%** |

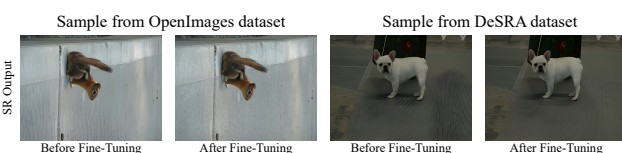

*Figure 8.* Examples of SR artifacts removed after fine-tuning with the proposed metric. Zoom in to see in full resolution.

Figure 8 demonstrates the resulting SR artifact reduction.

uct of Xie et al. (2023); the results are in the last column. We tested LDL at two lower thresholds, which increased the total masks found, but they mainly captured non-prominent artifacts, yielding worse combined score.

### 5.5. Fine-tuning SR models to reduce artifacts

Following the methodology from Xie et al. (2023), we fine-tune SR models to reduce artifacts using artifact-detection methods, and compare all SR-detector combinations. Fine-tuning optimizes the pixel-wise MSE loss between the SR model's output and an artificial GT image, constructed by replacing the artifact regions on the SR model's output with output from RLFN. The artifact regions are the binarized output masks of a given artifact-detection method.

We used the DeSRA dataset for both training and testing, with the split determined by the target SR model (RealESR-GAN, LDL, or SwinIR). For a given model, images with an artifact mask for that model were reserved for testing, while the remaining images were used for fine-tuning. This setup enabled us to evaluate IoU, artifact removal, and addition against GT masks on the held-out test set. Each model was trained on roughly 300 images and tested on about 200.

We measure metrics from Xie et al. (2023): $\Delta$IoU (average reduction of IoU with GT artifact mask), and Add$_{img}$ and Rem$_{img}$ (image-wise artifact removal and addition rates). As Table 6 shows, our method demonstrates the most consistent performance. See Section C for additional experiments.

## 6. Conclusion

We addressed the challenge of visual artifacts in single-image super-resolution by introducing the concept of *prominence*—characterizing artifacts by their perceptual impact rather than binary masks. Our crowdsourced study revealed that nearly half the artifacts in the existing DeSRA dataset go unnoticed by most viewers, validating this perspective. We contribute a new dataset of 1302 artifact examples with prominence annotations from 11 contemporary SISR methods, along with prominence labels for all 593 DeSRA artifacts. Building on this data, we developed a lightweight method that outputs prominence heatmaps and demonstrated its utility for evaluating SR models and fine-tuning them for artifact suppression.

The implications of our work extend beyond SISR. Our evaluation on JPEG AI artifacts in Section H suggests that prominence modeling applies to other restoration tasks. Further, prominence-aware metrics could guide SR research toward structured regions where artifacts are most visible.

We acknowledge limitations: the artifact masks in our dataset are approximate, as delineating exact artifact boundaries is ambiguous even for human annotators. Additionally, our pseudo-GT approach can produce false positives when the lightweight SR model fails to reconstruct fine textures.

Future work could extend prominence modeling to video super-resolution, addressing temporal artifacts such as flickering. Additionally, new higher-capacity models like SUPIR start to produce semantic artifacts (e.g., object replacement) rather than simple texture distortions—a shift that warrants further investigation.

## Impact Statement

This paper presents work whose goal is to advance the field of image and video super-resolution. We aim to improve the robustness of SR methods to artifacts, making the output images appear more realistic with no disturbing details. Meanwhile, we make no attempt to prevent SR methods from generating natural-looking but incorrect details. This may be of critical importance to certain applications like face or license plate recognition, but falls outside the scope of our work.

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

## A. Input feature details

This section provides implementation details for the input features described in Section 4.1.

### A.1. Structure Similarity Map (ssm_jup)

Our *ssm_jup* feature adapts the small-color-artifact detector from Tsereh et al. (2024), which is itself based on LDL (Liang et al., 2022). Given a reference image $I_{\text{ref}}$ (the original HR or pseudo-GT), the SR output $I_{\text{SR}}$, and a bicubic-upscaled baseline $I_{\text{bic}}$, we compute scaled residual-variance maps for both SR and bicubic outputs.

First, we compute the absolute residual summed across channels:

$$R_x^C(i,j) = \sum_{c \in C} \left| I_x^c(i,j) - I_{\text{ref}}^c(i,j) \right|, \quad x \in \{\text{SR}, \text{bic}\}. \tag{3}$$

We then compute local variance within an $n \times n$ window and scale by a global factor:

$$M_x^C(i,j) = var\left(R_x^C\left(i - \frac{n-1}{2} : i + \frac{n-1}{2}, j - \frac{n-1}{2} : j + \frac{n-1}{2}\right)\right), \tag{4}$$

$$S_x^C(i,j) = var\left(R_x^C\right)^{1/5} \cdot M_x^C(i,j), \tag{5}$$

where $n = 33$.

The key modification from Tsereh et al. (2024) is in the choice of color channels $C$: the original method computes separate maps on chrominance channels (UV from YUV and ab from Lab color spaces) and intersects the thresholded results to detect color artifacts. We instead operate on all three RGB channels ($C = \{R, G, B\}$), which enables detection of luminance-correlated texture distortions that are common in SR artifacts but would be missed by chrominance-only analysis.

The final feature is the smoothed difference between the SR and bicubic maps:

$$\text{ssm\_jup} = G_\sigma * S_{\text{SR}} - G_\sigma * S_{\text{bic}}, \tag{6}$$

where $G_\sigma$ denotes a Gaussian kernel with $\sigma = 33$.

### A.2. Block-wise Distortion (bd_jup)

The *bd_jup* feature combines block-wise LPIPS (Zhang et al., 2018) and ERQA (Kirillova et al., 2022) scores. LPIPS is computed on $32 \times 32$ blocks with stride 16; ERQA uses $8 \times 8$ blocks with no overlap. Since ERQA measures edge-preservation quality (higher is better), we invert it to obtain a distortion score. The final feature is:

$$\text{bd\_jup} = 0.6 \cdot \text{LPIPS} + 0.4 \cdot (1 - \text{ERQA}). \tag{7}$$

## B. Crowdsourced annotation dispersion analysis

Our crowdsourced prominence-annotation work, described in Section 3.1, involved 30 participants ranking every image separately. This appendix provides our motivation for choosing this number by analyzing the answer dispersion.

We took 11 SR-upscaled images with artifacts of varying intensity and conducted crowdsourced prominence annotation following the same procedure, but with a higher participant count: every image underwent ranking by 264 people. Next, we performed a bootstrap analysis on the votes. For each assessor count $k$ from 1 to 100, the analysis randomly sampled $k$ votes with replacement and computed the prominence from these votes. This procedure repeated $n$=1000 times; we then computed 95% confidence intervals for each assessor count $k$.

Figure 9 shows these confidence intervals for two sample images: one with a highly prominent artifact and another with a barely prominent artifact. In cases with few assessors (1–5), the confidence interval frequently spans the whole prominence range from 0% to 100%, meaning any given 5 assessors may all state that an artifact is present or absent. This is especially true for unclear cases at around 50% prominence. As the assessor count grows, the confidence interval shrinks, reaching approximately ±10% at 100 assessors.

For the rest of our annotation process we chose an assessor count of 30 as a reasonable compromise between the confidence of the result (±20%) and the time/cost of using many assessors.

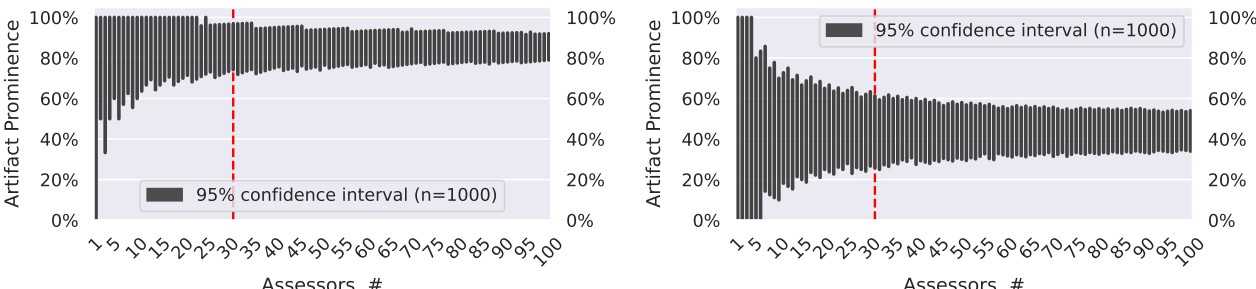

*Figure 9.* Bootstrap-analysis results for an image with a highly prominent artifact (left) and barely prominent artifact (right). Red line indicates our chosen assessor count of 30.

*Table 8.* Results of fine-tuning SR models on artificial GT constructed with different methods with and without dilation. The top half of the table matches the table in the main paper text.

| Target SR | LDL (Liang et al., 2022) | | | | | RealESRGAN (Wang et al., 2021b) | | | | | SwinIR (Liang et al., 2021) | | | | |
|---|---|---|---|---|---|---|---|---|---|---|---|---|---|---|---|
| Method | $\Delta$IoU$^\uparrow$ | Add$_{img}^\downarrow$ | Rem$_{img}^\uparrow$ | Add$_{pix}^\downarrow$ | Rem$_{pix}^\uparrow$ | $\Delta$IoU$^\uparrow$ | Add$_{img}^\downarrow$ | Rem$_{img}^\uparrow$ | Add$_{pix}^\downarrow$ | Rem$_{pix}^\uparrow$ | $\Delta$IoU$^\uparrow$ | Add$_{img}^\downarrow$ | Rem$_{img}^\uparrow$ | Add$_{pix}^\downarrow$ | Rem$_{pix}^\uparrow$ |
| **Dilation = True** | | | | | | | | | | | | | | | |
| DISTS (t=0.25) | 27.00 | 33.51 | 11.70 | **0.03** | 93.27 | 33.47 | 32.66 | 15.58 | 0.03 | 96.49 | **15.23** | 26.47 | 13.73 | 0.02 | 93.38 |
| LDL (t=0.005) | 17.26 | 90.43 | 6.91 | 0.25 | **97.95** | 18.32 | 88.94 | 11.56 | 0.13 | **98.40** | 4.22 | 77.94 | 21.08 | 0.04 | **98.03** |
| LPIPS (t=0.25) | 25.18 | **15.43** | 19.68 | 0.15 | 82.50 | 33.66 | **8.54** | 34.17 | **0.01** | 90.56 | 11.21 | **2.94** | 53.43 | **0.00** | 83.55 |
| ERQA (t=0.55) | 0.67 | 100.00 | 0.00 | 0.53 | 35.34 | 1.22 | 99.50 | 0.00 | 0.56 | 36.46 | 0.19 | 98.04 | 0.00 | 0.20 | 40.95 |
| DeSRA | 29.18 | 54.26 | 25.00 | 0.58 | 71.02 | 33.66 | 34.17 | **56.78** | 0.25 | 80.06 | 8.98 | 32.84 | 28.43 | 0.12 | 51.69 |
| **Ours** (t=0.3) | **34.71** | 20.74 | **45.21** | **0.03** | 97.20 | **38.01** | 14.57 | 55.78 | 0.03 | 98.09 | 11.86 | 6.37 | **57.35** | **0.00** | 91.95 |
| **Dilation = False** | | | | | | | | | | | | | | | |
| DISTS (t=0.25) | 24.97 | 52.13 | 2.13 | **0.05** | 88.22 | 32.93 | 47.24 | 7.04 | **0.04** | **94.17** | **14.58** | 25.98 | 7.35 | 0.01 | **91.03** |
| LDL (t=0.005) | 12.47 | 100.00 | 0.00 | 0.36 | 77.24 | 15.06 | 99.50 | 0.50 | 0.38 | 82.11 | 3.65 | 98.04 | 0.98 | 0.07 | 79.31 |
| LPIPS (t=0.25) | 23.65 | **18.09** | 17.02 | 0.16 | 78.20 | 25.13 | **15.08** | 32.20 | **0.04** | 87.38 | 10.90 | **3.92** | **47.06** | **0.00** | 82.39 |
| ERQA (t=0.55) | 0.34 | 98.94 | 0.00 | 0.37 | 19.04 | 0.45 | 99.50 | 0.00 | 0.51 | 20.40 | 0.06 | 97.06 | 0.00 | 0.12 | 24.18 |
| DeSRA | 27.59 | 57.45 | **21.81** | 0.53 | 68.33 | 33.23 | 39.20 | **50.25** | 0.31 | 77.36 | 8.47 | 35.29 | 25.98 | 0.15 | 49.89 |
| **Ours** (t=0.3) | **31.63** | 45.21 | 12.77 | 0.14 | **88.83** | **36.83** | 48.24 | 23.12 | 0.12 | 93.49 | 11.01 | 24.02 | 30.88 | **0.00** | 87.50 |

# C. SR fine-tuning scoring and dilation details

Section 5.5 overviews our fine-tuning process to reduce artifacts of existing SR models. Here, we provide more details on the pipeline and on the scoring process.

The input low-resolution image is upscaled with the target SR and RLFN models, and the results are passed to the artifact metric. Then, the regions on the target-SR output where the artifacts were detected are replaced with regions from the RLFN output. The resulting artificial GT image is then used as a target to fine-tune the target-SR model.

Table 6 in the main section uses several scores to compare fine-tuning results. Below are the detailed descriptions of these scores. In the following formulas, $A$ represents the set of pixels detected by a metric after fine-tuning, $B$ represents the set of pixels detected by a metric before fine-tuning, and $GT$ represents the set of pixels belonging to the ground-truth artifact mask annotations (from the DeSRA dataset).

- $\Delta$IoU. Average reduction of IoU with GT artifact mask: $\frac{|B \cap GT|}{|B \cup GT|} - \frac{|A \cap GT|}{|A \cup GT|}$.

- Rem$_{img}$ and Add$_{img}$, image-wise removal and addition rates. The ratio of images where $(|A \cap B| = 0) \wedge (|B| \neq 0)$ determines whether the artifact was removed, and the ratio where $|A \cup B| > |B|$ determines whether a new artifact was introduced.

- Rem$_{img}$: the image-wise removal rate. It represents the percentage of images in test set in which fine-tuning removed artifact regions previously detected. The condition for determining whether an artifact was removed is $(|A \cap B| = 0) \wedge (|B| \neq 0)$. We add a new condition $(|B| \neq 0)$ to prevent metrics gaining an increase in this score by introducing new artifacts on previously clear images.

- Add$_{pix}$. Pixel-wise addition rate; represents mean percentage of pixels in new artifact regions that resulted from fine-tuning: $|A \cap \overline{B}| / |\overline{B}|$.

*Table 9.* Results of fine-tuning SR models on artificial GT constructed with different methods (absolute values).

| Target SR | LDL (Liang et al., 2022) | | | | RealESRGAN (Wang et al., 2021b) | | | | SwinIR (Liang et al., 2021) | | | |
|---|---|---|---|---|---|---|---|---|---|---|---|---|
| **Method** | IoU before | IoU after | PixFrac before | PixFrac after | IoU before | IoU after | PixFrac before | PixFrac after | IoU before | IoU after | PixFrac before | PixFrac after |
| DISTS (t=0.25) | 29.51 | 2.51 | 24.69 | 2.00 | 34.30 | 0.83 | 9.44 | 0.30 | 16.53 | 1.30 | 20.73 | 1.92 |
| LDL (t=0.005) | 18.08 | 0.82 | 8.20 | 0.47 | 18.88 | 0.56 | 3.10 | 0.18 | 4.27 | 0.05 | 3.90 | 0.21 |
| LPIPS (t=0.25) | 30.62 | 5.44 | 8.48 | 2.29 | 36.37 | 2.71 | 1.50 | 0.18 | 12.20 | 0.99 | 5.02 | 0.96 |
| ERQA (t=0.55) | 4.64 | 3.97 | 4.95 | 3.96 | 5.94 | 4.72 | 5.17 | 3.04 | 1.00 | 0.81 | 2.52 | 1.73 |
| DeSRA | 35.73 | 6.55 | 9.04 | 4.68 | 35.35 | 1.69 | 4.68 | 0.36 | 13.06 | 4.08 | 5.26 | 2.73 |
| **Ours** (t=0.3) | 35.86 | 1.15 | 12.57 | 0.99 | 38.31 | 0.30 | 2.28 | 0.18 | 12.06 | 0.20 | 5.77 | 0.42 |

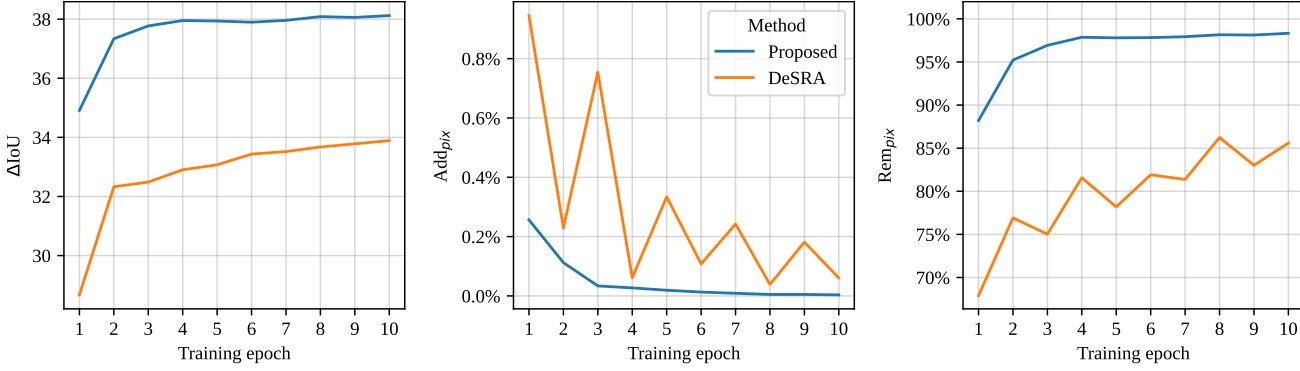

*Figure 10.* Evolution of metric values over 10 epochs of RealESRGAN fine-tuning with DeSRA compared to the proposed method.

- $Rem_{pix}$. Pixel-wise removal rate; represents mean percentage of pixels that were previously classified as artifacts and that go undetected after fine-tuning: $\left|\overline{A} \cap B\right| / |B|$.

We report metrics after five epochs of fine-tuning. This is sufficient to saturate the training; Figure 10 shows metric evolution for ten fine-tuning epochs.

We mentioned that we dilated the masks, which improved the fine-tuning quality. We observed that the fine-tuning artificial GT image obtained by masking the upscaled image with the detected artifact mask sometimes produces a poor quality image. In cases when the artifact mask is too tight or noisy, some parts of the artifact region are still present on artificial GT, which may decrease the quality of fine-tuned models. Both of these cases can be solved by dilating the artifact mask before constructing the artificial GT.

Table 8 shows fine-tuning results for all metrics when using dilation (this part of the table matches Table 6 from the main paper), and when using unmodified artifact masks. The proposed dilation step improves the results across all scores and metrics.

Finally, Table 9 provides absolute values for IoU and for the fraction of pixels detected as artifacts, before and after the fine-tuning procedure. ΔIoU in the main tables corresponds to the difference between "IoU before" and "IoU after".

## D. Subjective quality of images with artifacts

When a super-resolved image contains an artifact, its subjective quality typically decreases, as reflected in assessor scores. This observation shows the importance of artifact detection, since artifact-free images tend to be more visually pleasant than those with artifacts.

To validate this claim, we conducted a side-by-side subjective comparison using images from our dataset. Participants viewed pairs of super-resolved images and were asked to select from each pair the one they preferred. A total of 842 people took part in our subjective study. Calculation of the final scores used the Bradley-Terry model and included 16,720 votes.

We then analyzed the correlation between these subjective scores and $(1 - p)$, where $p$ is the prominence score of the artifact on the image. As Table 10 shows, the correlation was positive in most cases, supporting our hypothesis that artifact-free images tend to receive higher subjective scores.

*Table 10.* Correlation between subjective scores and $(1 - p)$, where $p$ is the prominence score. We calculated subjective scores independently for each group of images that share the same input low-resolution image.

| Group | Pearson | Spearman |
|---|---|---|
| image1 | 1.00 | 1.00 |
| image2 | 1.00 | 0.95 |
| image3 | 0.93 | 0.74 |
| image4 | 0.84 | 0.95 |
| image5 | 0.76 | 0.40 |
| image6 | 0.47 | 0.00 |
| image7 | 0.31 | 0.20 |
| image8 | 0.23 | 0.40 |
| image9 | 0.22 | 0.32 |
| image10 | 0.21 | 0.40 |
| image11 | -0.63 | -0.80 |
| image12 | -0.76 | -0.95 |

*Table 11.* Crowd-sourced prominence across SR models on 6 datasets.

| SR | Type | Masks Found | Mean Prominence | Conf. Masks Found |
|---|---|---|---|---|
| SeeSR | Diffusion | 61 | **7.15%** | 0 |
| ResShift | Diffusion | 60 | 7.20% | 0 |
| SinSR | Diffusion | 60 | 9.19% | 2 |
| HAT-L | Transformer | 50 | 9.60% | 1 |
| DRCT | Transformer | 50 | 13.33% | 0 |
| SwinIR | Transformer | 60 | 17.12% | 4 |
| RealESRGAN | CNN | 60 | 17.46% | 5 |
| SUPIR | Diffusion | 62 | 17.56% | 6 |

*Table 12.* Crowd-sourced prominence across artifact detection methods on 6 datasets.

| Method | Masks Found | Mean Prominence | Conf. Masks Found | Comb. Score |
|---|---|---|---|---|
| ssm_jup (t=0.15) | 80 | 7.42% | 1 | 0.07 |
| bd_jup (t=0.1) | 80 | 7.34% | 2 | 0.15 |
| LDL (t=0.005) | 80 | 9.46% | 2 | 0.19 |
| DeSRA | 74 | 12.21% | 3 | 0.37 |
| **Ours** (t=0.3) | 70 | 17.85% | 8 | 1.43 |
| DISTS (t=0.25) | 76 | **18.03%** | 9 | 1.62 |

## E. Artifact examples and failure cases

Figure 11 shows examples of prominent artifacts detected by our proposed method across various SR models (Yu et al., 2024; Wang et al., 2021b; 2024a; Liang et al., 2021; Wang et al., 2021a). Each example is annotated with the binary artifact mask and subjective prominence.

Figure 12 shows examples of false detections by our proposed method across SR models (Chen et al., 2023; Hsu et al., 2024; Yue et al., 2023; Wang et al., 2024b; Cai et al., 2019). We observed the following failure cases:

- Distortions on natural, unstructured objects, like ground, grass, or trees, that are not very prominent to human observers.

- Accurate restoration of fine textures such as fur, nylon, or mesh grille. False detections can happen on these when the lightweight SR (in our case, RLFN (Kong et al., 2022)) fails to produce a sharp upscaling of the texture, leading the metrics to see a discrepancy to the target SR and mark it as an artifact. Using an accurately-restored reference removes those false detections as Figure 13 shows.

Existing methods also suffer from these failure cases; indeed, they account for most of the low-prominence detections from our subjective evaluation described in Section 5.4.

## F. Subjective evaluation on additional SR datasets

We conduct an additional subjective evaluation on 6 widely known image datasets (Martin et al., 2001; Wang et al., 2018; Dong & Loy, 2016; Bevilacqua et al., 2012; Yang et al., 2010), following the setup described in Section 5.4. In total, this evaluation used 420 source images, each processed by 8 SR models.

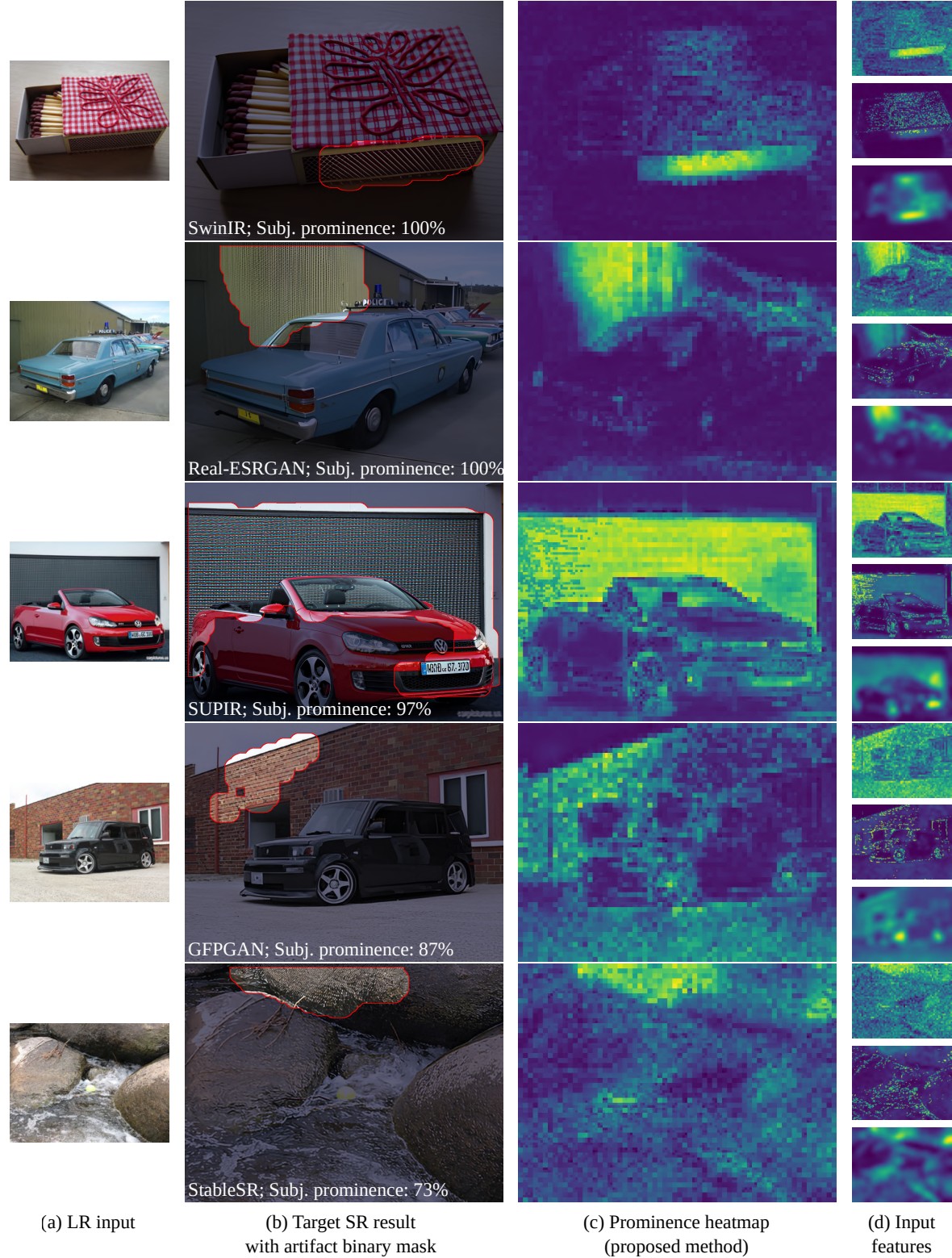

(a) LR input

(b) Target SR result
with artifact binary mask

(c) Prominence heatmap
(proposed method)

(d) Input
features

*Figure 11.* Example artifacts detected by the proposed method. (a): low-resolution input image; (b): target SR result with annotated output artifact mask; (c): artifact prominence heatmap predicted by our method; (d): our input features described in Sec. 4.1, top to bottom: DISTS, bd_jup, ssm_jup.

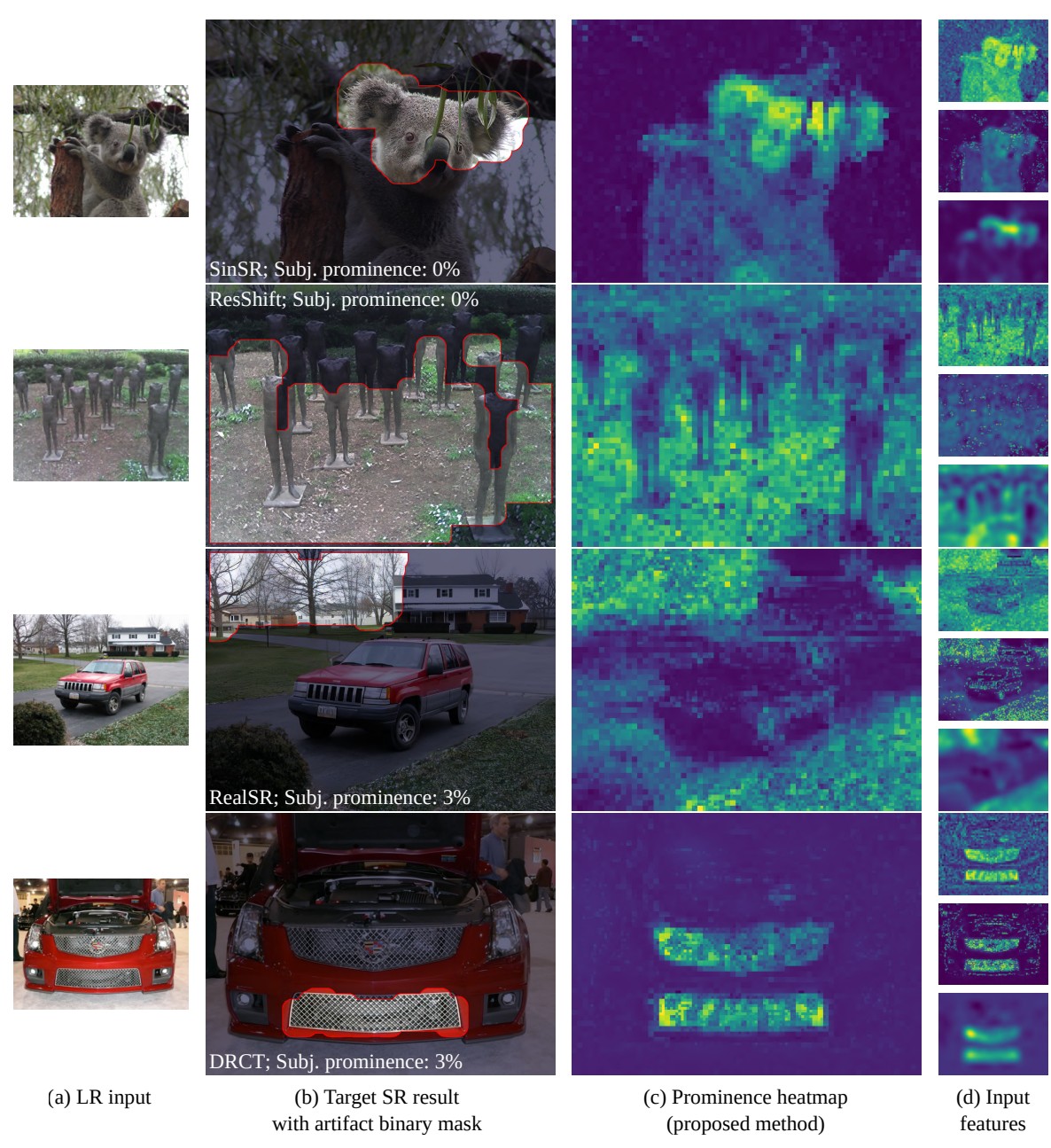

(a) LR input | (b) Target SR result with artifact binary mask | (c) Prominence heatmap (proposed method) | (d) Input features

*Figure 12.* Example false detections by the proposed method. (a): low-resolution input image; (b): target SR result with annotated output artifact mask; (c): artifact prominence heatmap predicted by our method; (d): our input features described in Sec. 4.1, top to bottom: DISTS, bd_jup, ssm_jup.

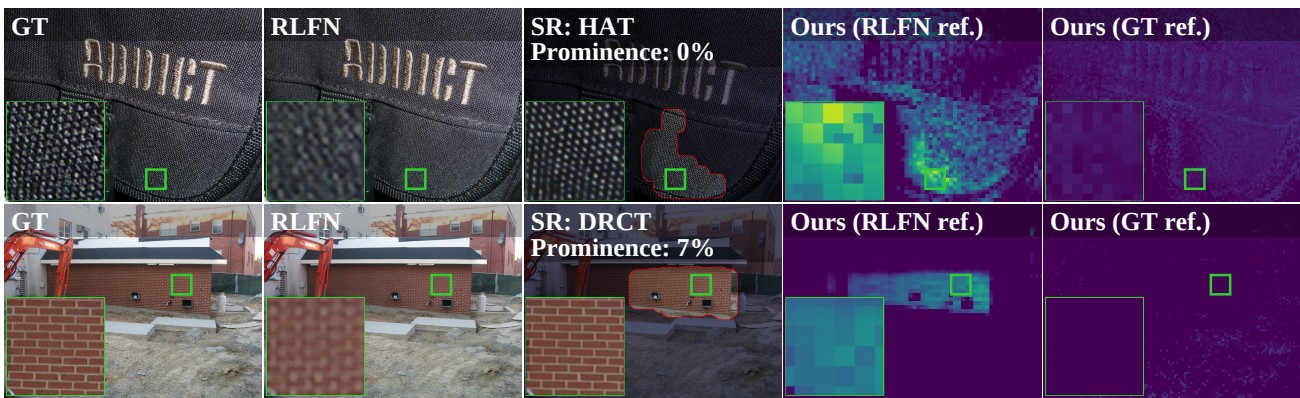

*Figure 13.* Example false detections by the proposed method due to inaccurate restoration from pseudo-GT lightweight SR (RLFN). Rightmost column shows that the false detection disappears when an accurate restoration is used as reference instead of RLFN.

*Table 13.* Results on the full JPEG AI edge artifact dataset.

| Method | $Prec^{pr}$ | $Rec^{pr}$ | F1-score | PR-AUC |
|---|---|---|---|---|
| DISTS (t=0.25) | 0.0655 | 0.1783 | 0.0958 | **0.0331** |
| bd_jup (t=0.1) | 0.0599 | 0.4261 | 0.1050 | 0.0330 |
| ssm_jup (t=0.2) | 0.0784 | 0.1083 | 0.0910 | 0.0324 |
| DeSRA | 0.0784 | 0.0518 | 0.0623 | 0.0218 |
| **Ours (t=0.15)** | 0.0742 | 0.1907 | **0.1068** | 0.0295 |
| **Ours (t=0.3)** | 0.0835 | 0.0883 | 0.0858 | 0.0295 |

*Table 14.* Results on the prominent subset of the JPEG AI edge artifact dataset.

| Method | Precision | Recall | Prec * Rec | IoU | PR-AUC |
|---|---|---|---|---|---|
| DISTS (t=0.25) | 0.0779 | 0.2339 | 0.0182 | 0.0699 | 0.0444 |
| bd_jup (t=0.1) | 0.0467 | 0.6765 | **0.0316** | **0.0912** | **0.0554** |
| ssm_jup (t=0.2) | 0.1051 | 0.1322 | 0.0139 | 0.0584 | 0.0479 |
| DeSRA | 0.1145 | 0.0481 | 0.0055 | 0.0313 | 0.0414 |
| **Ours (t=0.15)** | 0.0778 | 0.2667 | 0.0208 | 0.0825 | 0.0526 |
| **Ours (t=0.3)** | 0.1179 | 0.0951 | 0.0112 | 0.0520 | 0.0526 |

Tables 11 and 12 show the results, grouped by SR models and by artifact detection methods, respectively. Interestingly, SR models show much better artifact robustness than in our main comparison in Section 5.4, likely because these datasets are commonly used for SR training and evaluation. Our proposed method falls one confident artifact short of DISTS, but otherwise shows competitive results.

## G. F1-score bounds analysis

The theoretical bounds of F1-score are [-0.3, 0.7], governed by the penalty term $\kappa = 0.3$. The practical range of scores achieved by any non-trivial method depends on the dataset's GT prominence distribution. Our dataset exhibits the following prominence statistics: mean = 0.40, std. dev. = 0.30, with 53% of masks having a value >0.3. Compared to the DeSRA dataset (mean = 0.49, std. dev. = 0.26, 71% masks >0.3), our labels are skewed toward less prominent regions. This difference in dataset bias results in generally higher absolute metric scores on DeSRA.

## H. Objective evaluation on the learning-based compression task

Learning-based image compression has seen a lot of research attention recently, especially as efforts focused on finalizing the JPEG AI compression standard (Ascenso et al., 2023). JPEG AI promises considerable bitrate savings compared to traditional compression. Unfortunately, as Tsereh et al. (2024) show, JPEG AI is also susceptible to the neural artifacts problem, not dissimilar to learning-based super-resolution.

In order to evaluate the transferability of our promised method to other domains, we conduct an objective evaluation on the JPEG AI edge artifact examples dataset collected by Tsereh et al., following our methodology described in Section 5.1. Tables 13 and 14 show the results.

Our proposed method achieves the best F1-score on the full set, and second-best IoU and PR-AUC on the prominent subset, indicating good transferability across domains. We expect the performance to increase further if the proposed method was fine-tuned on artifact examples specific to JPEG AI.

*Table 15.* Ablation results on the full proposed dataset.

| | Original HR | | | | SPAN | | | | RLFN | | | |
|---|---|---|---|---|---|---|---|---|---|---|---|---|
| **Method** | $Prec^{pr}$ | $Rec^{pr}$ | **F1-score** | **PR-AUC** | $Prec^{pr}$ | $Rec^{pr}$ | **F1-score** | **PR-AUC** | $Prec^{pr}$ | $Rec^{pr}$ | **F1-score** | **PR-AUC** |
| CNN (t=0.45) | 0.1204 | 0.0526 | **0.0732** | **0.0122** | 0.1023 | 0.0379 | 0.0553 | **0.0083** | 0.102 | 0.0348 | 0.0520 | 0.0082 |
| w/o bd_jup (t=0.45) | 0.0621 | 0.0485 | 0.0545 | 0.0011 | 0.0454 | 0.0267 | 0.0336 | 0.0008 | 0.0456 | 0.0254 | 0.0327 | 0.0008 |
| w/o ssm_jup (t=0.25) | 0.0572 | 0.0477 | 0.0520 | 0.0029 | 0.0948 | 0.0406 | 0.0569 | 0.0066 | 0.1036 | 0.0377 | 0.0553 | 0.0075 |
| w/o DISTS (t=0.2) | 0.0147 | 0.0232 | 0.0180 | 0.0006 | 0.0177 | 0.0209 | 0.0191 | 0.0010 | 0.0095 | 0.0110 | 0.0102 | 0.0009 |
| HR+SPAN+RLFN (t=0.35) | 0.0604 | 0.0470 | 0.0528 | 0.0013 | 0.0588 | 0.0330 | 0.0423 | 0.0022 | 0.0628 | 0.0312 | 0.0417 | 0.0028 |
| HR+RLFN (t=0.3) | 0.0499 | 0.0501 | 0.0500 | 0.0039 | 0.0651 | 0.0422 | 0.0512 | 0.0081 | 0.0717 | 0.0413 | 0.0524 | **0.0086** |
| Just RLFN (t=0.4) | 0.0273 | 0.0322 | 0.0296 | 0.0003 | 0.0885 | 0.0502 | **0.0641** | 0.0013 | 0.1241 | 0.0493 | **0.0706** | 0.0031 |
| GT-area-only (t=0.35) | 0.0224 | 0.0243 | 0.0233 | 0.0107 | 0.0130 | 0.0096 | 0.0110 | 0.0070 | 0.0130 | 0.0090 | 0.0106 | 0.0054 |
| Weighted loss (t=0.4) | 0.0775 | 0.0413 | 0.0539 | 0.0094 | 0.0485 | 0.0195 | 0.0278 | 0.0061 | 0.0539 | 0.0201 | 0.0293 | 0.0061 |
| **Ours (t=0.15)** | 0.0317 | 0.0402 | 0.0355 | 0.0121 | 0.0335 | 0.0315 | 0.0325 | 0.0075 | 0.0338 | 0.0290 | 0.0312 | 0.0075 |
| **Ours (t=0.3)** | 0.0762 | 0.0441 | 0.0559 | | 0.0503 | 0.0224 | 0.0310 | | 0.0581 | 0.0235 | 0.0334 | |

*Table 16.* Ablation results on the full DeSRA dataset.

| | MSE-SR | | | | SPAN | | | | RLFN | | | |
|---|---|---|---|---|---|---|---|---|---|---|---|---|
| **Method** | $Prec^{pr}$ | $Rec^{pr}$ | **F1-score** | **PR-AUC** | $Prec^{pr}$ | $Rec^{pr}$ | **F1-score** | **PR-AUC** | $Prec^{pr}$ | $Rec^{pr}$ | **F1-score** | **PR-AUC** |
| CNN (t=0.45) | 0.1999 | 0.1750 | 0.1866 | 0.0536 | 0.1656 | 0.1880 | **0.1761** | 0.0452 | 0.1932 | 0.1792 | 0.1860 | 0.0528 |
| w/o bd_jup (t=0.45) | 0.1936 | 0.1846 | 0.1890 | 0.0603 | 0.1059 | 0.2086 | 0.1405 | 0.0375 | 0.1871 | 0.1867 | 0.1869 | 0.0589 |
| w/o ssm_jup (t=0.25) | 0.1718 | 0.1557 | 0.1634 | 0.0467 | 0.0912 | 0.1898 | 0.1232 | 0.0242 | 0.1716 | 0.1606 | 0.1660 | 0.0488 |
| w/o DISTS (t=0.2) | 0.1308 | 0.1970 | 0.1572 | 0.0346 | 0.0814 | 0.2087 | 0.1171 | 0.0188 | 0.1266 | 0.1986 | 0.1547 | 0.0349 |
| HR+SPAN+RLFN (t=0.35) | 0.1944 | 0.1821 | 0.1880 | 0.0612 | 0.1016 | 0.2097 | 0.1369 | 0.0380 | 0.1893 | 0.1842 | 0.1868 | 0.0594 |
| HR+RLFN (t=0.3) | 0.1686 | 0.1890 | 0.1782 | 0.0611 | 0.0869 | 0.2157 | 0.1239 | 0.0345 | 0.1647 | 0.1910 | 0.1769 | 0.0609 |
| Just RLFN (t=0.4) | 0.1415 | 0.1724 | 0.1554 | 0.0405 | 0.0604 | 0.1975 | 0.0925 | 0.0167 | 0.1416 | 0.1755 | 0.1568 | 0.0394 |
| GT-area-only (t=0.35) | 0.1870 | 0.1778 | 0.1823 | 0.0500 | 0.1355 | 0.1922 | 0.1589 | 0.0372 | 0.1781 | 0.1818 | 0.1799 | 0.0521 |
| Weighted loss (t=0.4) | 0.2282 | 0.1629 | 0.1901 | 0.0575 | 0.1674 | 0.1755 | 0.1714 | **0.0481** | 0.2183 | 0.1671 | 0.1893 | 0.0558 |
| **Ours (t=0.15)** | 0.1565 | 0.2063 | 0.1780 | **0.0616** | 0.0849 | 0.2262 | 0.1235 | 0.0398 | 0.1500 | 0.2064 | 0.1737 | **0.0605** |
| **Ours (t=0.3)** | 0.2170 | 0.1701 | **0.1907** | | 0.1271 | 0.1955 | 0.1540 | | 0.2117 | 0.1727 | **0.1902** | |

# I. Additional experiments

This appendix describes our additional experiments and ablation studies. For all experiments, we measure objective metrics as we described in Section 5.1 of the main paper. Tables 15 and 17 show the results on our proposed dataset, and Tables 16 and 18 show the results on the DeSRA (Xie et al., 2023) dataset.

The following sections describe these model variations in detail.

## I.1. Input-feature ablation

As described in Section 4.2, our proposed method takes as input three features: DISTS, bd_jup, and ssm_jup. We train and evaluate three variations of our method, excluding each of these features. These variants are marked "w/o [feature]" in the results tables.

Removing DISTS and ssm_jup results in the heaviest performance drops to our method. This is consistent with the high scores of DISTS alone in our evaluations: this metric is quite capable for detecting artifacts. Removing bd_jup, on the other hand, gives very similar performance to our full proposed method on most evaluations. However, the proposed method still generally outperforms this variant.

## I.2. Training on different pseudo-GT

We mention in Section 5.2 that we use RLFN as the pseudo-GT input for all experiments in the paper, unless noted otherwise, since it showed better performance compared to SPAN. It makes sense then that we should also train our proposed method with RLFN passed as the pseudo-GT input. However, this was not the case. We trained our final method using the original high-resolution input, despite evaluating it with the RLFN pseudo-GT.

The results tables show scores for our proposed method's checkpoints trained using RLFN as pseudo-GT (denoted "Just RLFN"), as well as original high-resolution mixed with RLFN (denoted "HR+RLFN"), and original high-resolution mixed with RLFN and SPAN (denoted "HR+SPAN+RLFN").

*Table 17.* Ablation results on the prominent subset of the proposed dataset.

| | Original HR | | | | | SPAN | | | | | RLFN | | | | |
|---|---|---|---|---|---|---|---|---|---|---|---|---|---|---|---|
| Method | Precision | Recall | Prec * Rec | IoU | PR-AUC | Precision | Recall | Prec * Rec | IoU | PR-AUC | Precision | Recall | Prec * Rec | IoU | PR-AUC |
| CNN (t=0.45) | 0.4560 | 0.4836 | 0.2206 | 0.3510 | 0.4495 | 0.5263 | 0.320 | 0.1684 | 0.2375 | **0.4034** | 0.5320 | 0.3127 | 0.1664 | 0.2338 | **0.4072** |
| w/o bd_jup (t=0.45) | 0.3614 | 0.6982 | **0.2523** | **0.3869** | 0.1758 | 0.4816 | 0.4182 | 0.2014 | 0.2694 | 0.2327 | 0.4830 | 0.4036 | 0.1949 | 0.2626 | 0.2225 |
| w/o ssm_jup (t=0.25) | 0.3454 | 0.6582 | 0.2273 | 0.3686 | 0.3762 | 0.4907 | 0.3527 | 0.1731 | 0.2666 | 0.3807 | 0.5174 | 0.2945 | 0.1524 | 0.2507 | 0.3780 |
| w/o DISTS (t=0.2) | 0.1693 | 0.7709 | 0.1305 | 0.2920 | 0.1717 | 0.1935 | 0.5818 | 0.1126 | 0.2471 | 0.2055 | 0.2108 | 0.5418 | 0.1142 | 0.2511 | 0.2065 |
| HR+SPAN+RLFN (t=0.35) | 0.3663 | 0.6873 | 0.2517 | 0.3842 | 0.3526 | 0.5000 | 0.4109 | 0.2055 | 0.2811 | 0.3317 | 0.4855 | 0.3855 | 0.1872 | 0.2702 | 0.3410 |
| HR+RLFN (t=0.3) | 0.2913 | 0.7491 | 0.2182 | 0.3801 | 0.4154 | 0.4060 | 0.4655 | 0.1890 | 0.2971 | 0.3962 | 0.4253 | 0.4291 | 0.1825 | 0.2856 | 0.4002 |
| Just RLFN (t=0.4) | 0.2260 | 0.6509 | 0.1471 | 0.3028 | 0.1202 | 0.3456 | 0.4982 | 0.1722 | 0.2881 | 0.2574 | 0.4124 | 0.3927 | 0.1619 | 0.2833 | 0.3077 |
| GT-area-only (t=0.35) | 0.2925 | 0.7018 | 0.2053 | 0.3314 | 0.3426 | 0.3524 | 0.4073 | 0.1435 | 0.2567 | 0.2793 | 0.3556 | 0.4073 | 0.1448 | 0.2452 | 0.2903 |
| Weighted loss (t=0.4) | 0.4712 | 0.5200 | 0.2450 | 0.3593 | 0.4714 | 0.5058 | 0.3091 | 0.1563 | 0.2223 | 0.3850 | 0.5111 | 0.3127 | 0.1598 | 0.2137 | 0.3711 |
| **Ours (t=0.15)** | 0.2447 | 0.8145 | 0.1993 | 0.3639 | **0.4756** | 0.3709 | 0.5636 | **0.2090** | **0.3018** | 0.3931 | 0.3890 | 0.5273 | **0.2051** | **0.2903** | 0.3829 |
| **Ours (t=0.3)** | 0.4357 | 0.5745 | 0.2503 | 0.3669 | | 0.5209 | 0.3345 | 0.1743 | 0.2357 | | 0.5138 | 0.3418 | 0.1756 | 0.2311 | |

*Table 18.* Ablation results on the prominent subset of the DeSRA dataset.

| | MSE-SR | | | | | SPAN | | | | | RLFN | | | | |
|---|---|---|---|---|---|---|---|---|---|---|---|---|---|---|---|
| Method | Precision | Recall | Prec * Rec | IoU | PR-AUC | Precision | Recall | Prec * Rec | IoU | PR-AUC | Precision | Recall | Prec * Rec | IoU | PR-AUC |
| CNN (t=0.45) | 0.5566 | 0.6116 | 0.3404 | 0.4999 | 0.6044 | 0.3575 | 0.6517 | **0.2330** | **0.4486** | 0.4072 | 0.5549 | 0.6645 | **0.3687** | 0.5132 | 0.6027 |
| w/o bd_jup (t=0.45) | 0.5195 | 0.6709 | 0.3485 | 0.5216 | 0.6057 | 0.2123 | 0.7897 | 0.1677 | 0.4238 | 0.3566 | 0.5043 | 0.7063 | 0.3562 | 0.5269 | 0.6059 |
| w/o ssm_jup (t=0.25) | 0.4964 | 0.5682 | 0.2821 | 0.4433 | 0.5226 | 0.2036 | 0.7352 | 0.1497 | 0.3712 | 0.2748 | 0.4961 | 0.6083 | 0.3018 | 0.4603 | 0.5426 |
| w/o DISTS (t=0.2) | 0.2165 | 0.7207 | 0.1560 | 0.4428 | 0.2688 | 0.1296 | 0.7753 | 0.1005 | 0.3436 | 0.1644 | 0.2017 | 0.7335 | 0.1480 | 0.4421 | 0.2619 |
| HR+SPAN+RLFN (t=0.35) | 0.5326 | 0.6565 | **0.3496** | 0.5162 | 0.6099 | 0.1958 | 0.7945 | 0.1555 | 0.4144 | 0.3544 | 0.5256 | 0.6902 | 0.3628 | 0.5248 | **0.6107** |
| HR+RLFN (t=0.3) | 0.4615 | 0.7159 | 0.3304 | 0.5192 | 0.5892 | 0.1547 | 0.8443 | 0.1306 | 0.3924 | 0.3363 | 0.4367 | 0.7528 | 0.3288 | 0.5263 | 0.5984 |
| Just RLFN (t=0.4) | 0.3399 | 0.6806 | 0.2313 | 0.4522 | 0.4292 | 0.1129 | 0.8138 | 0.0919 | 0.2946 | 0.1636 | 0.3216 | 0.7191 | 0.2312 | 0.4620 | 0.4355 |
| GT-area-only (t=0.35) | 0.4691 | 0.6019 | 0.2824 | 0.4697 | 0.3875 | 0.2741 | 0.6822 | 0.1870 | 0.4243 | 0.2490 | 0.4282 | 0.6453 | 0.2763 | 0.4756 | 0.3939 |
| Weighted loss (t=0.4) | 0.6520 | 0.5233 | 0.3412 | 0.4603 | 0.5975 | 0.3883 | 0.5746 | 0.2231 | 0.4220 | **0.4135** | 0.6078 | 0.5490 | 0.3336 | 0.4707 | 0.5830 |
| **Ours (t=0.15)** | 0.4121 | 0.7961 | 0.3281 | **0.5420** | **0.6104** | 0.1633 | 0.8973 | 0.1465 | 0.4010 | 0.3873 | 0.3353 | 0.8427 | 0.2825 | **0.5301** | 0.6030 |
| **Ours (t=0.3)** | 0.6088 | 0.5618 | 0.3420 | 0.4866 | | 0.2970 | 0.7175 | 0.2131 | 0.4374 | | 0.5543 | 0.6276 | 0.3479 | 0.5049 | |

We observe that training on pseudo-GT does tend to improve the results on our proposed dataset when using pseudo-GT for evaluation. However, when evaluating on the DeSRA dataset, training on the original high-resolution frames results in the best method performance, regardless of the pseudo-GT used during evaluation. We hypothesize that this may be because images in the DeSRA dataset have a higher resolution compared to our proposed dataset, making it easy for lightweight SRs to produce high quality pseudo-GT, which appears more similar to the original high-resolution frames.

### I.3. GT-area-only training

Our proposed dataset, described in Section 3, includes binary artifact masks and corresponding subjective prominence annotations. During training, our loss function, described in Section 5, moves the model towards predicting the subjective prominence value for pixels inside the binary artifact mask, and 0 (no artifact) outside the binary artifact mask.

However, we considered that this may not be entirely correct. During subjective annotation described in Section 3.1, we ask participants only if they see distortions inside the area denoted by the binary artifact mask. We dim the image to direct the participants' attention towards the masked area, and away from the other parts of the image. If the binary artifact mask missed an artifact elsewhere on the image, then the participants aren't expected to see and rank it. Effectively, we may not know the accurate artifact prominence value for regions outside the binary artifact mask, even though in practice those regions do not contain artifacts.

We conducted an experiment to account for this in training by disabling the loss component responsible for the region outside the binary artifact mask. This way, our loss function only considered the pixels within the artifact mask—for which we know the ground-truth subjective prominence value. Given our dataset's bias towards low-prominence samples, the model should still be able to learn when to predict the absence of an artifact.

The model from this training run is labeled "GT-area-only" in the results tables. We found that it failed to match the performance of the proposed model with the full loss function. Our hypothesis for this outcome is that the regions outside the binary artifact mask indeed contain no artifacts in most cases, and their loss function component helps the model learn to better localize artifacts on an image.

### I.4. Weighting loss to normalize training-set classes

As we pointed out in Section 3.3, our proposed dataset has a bias towards low-prominence samples. In this experiment, we tried to account for this bias in training by weighting the loss function according to the number of samples with the given

*Table 19.* Statistics across 4 proposed model checkpoints for full DeSRA and proposed datasets

| | | | F1-score | | PR-AUC | |
|---|---|---|---|---|---|---|
| **Dataset** | **Reference Input** | **Threshold** | **Mean** | **Std** | **Mean** | **Std** |
| Ours | Original HR | 0.15 | 0.0273 | 0.0086 | 0.0118 | 0.0009 |
| | | 0.30 | 0.0530 | 0.0126 | | |
| | RLFN | 0.15 | 0.0285 | 0.0023 | 0.0081 | 0.0004 |
| | | 0.30 | 0.0346 | 0.0074 | | |
| DeSRA | MSE-SR | 0.15 | 0.1644 | 0.0154 | 0.0608 | 0.0010 |
| | | 0.30 | 0.1866 | 0.0050 | | |
| | RLFN | 0.15 | 0.1586 | 0.0161 | 0.0595 | 0.0012 |
| | | 0.30 | 0.1837 | 0.0072 | | |

prominence (inversely proportional to the histogram on Figure 5).

The model from this training run is labeled "Weighted loss" in the results tables. It shows similar performance to the proposed model with uniformly weighted loss, and even overtakes it in specific scenarios, but the proposed model has better results overall. During training, we observed very similar validation curves between weighted and non-weighted loss.

### I.5. Variance across training runs

We trained our model across four random seeds. Table 19 demonstrates strong stability across seeds, as indicated by consistently low standard deviations relative to the mean values for both performance metrics. The PR AUC exhibits good stability, with fairly low standard deviations (0.0004–0.0012) across all dataset and ground-truth configurations. Similarly, the F1-score shows good reproducibility, on the both datasets. Overall, the low variance confirms that the reported mean performance metrics are highly reproducible and not artifacts of a single model initialization.

### I.6. Using a CNN instead of a per-pixel MLP

As we mentioned in Section 4.2, our proposed multilayer perceptron processes features for every pixel of the input image individually, which does not preclude it from using wider context, because our input features (DISTS, bd_jup, ssm_jup) themselves use wider context during computation. However, it is a reasonable assumption that the artifact prominence regressor itself may be able to extract additional useful contextual information from surrounding pixels, as the input features were not trained specifically for the artifact detection task.

We conducted an experiment replacing the multilayer perceptron with a small CNN. It consisted of five consecutive 3×3 convolution + ReLU + layer normalization residual blocks, each with 8 depth channels.

This model is labeled "CNN" in the results tables. We tried both passing just the original three features as input, and the original three features together with the normalized R, G, and B color channels of the input image. Both of these variants netted similar results, which failed to improve upon our proposed multilayer perceptron model.

### I.7. Using a random forest instead of MLP

We trained several random-forest models on the features described in Section 4.1 using different sets of hyperparameters: the number of estimators varied between 8, 16, and 32, and the maximum tree depth was 2, 4, 6, 8, 10, or 12. We used the random-forest implementation from the XGBoost library. All other hyperparameters except `n_estimators` and `max_depth` remain their default values. Empirically, we chose 0.05 as the best threshold for all trained random-forest models.

Our comparison used the test set and the $F1\text{-}score^{pr}$ metric. Table 20 shows the prediction scores of the trained models compared with our proposed method. Note that the random-forest architecture fails to achieve higher quality than DeSRA or our multilayer-perceptron architecture.

The experiments also investigated other training-data-preparation approaches, but they failed to exhibit strong increases in

*Table 20.* $F1$-$score^{pr}$ metric for random-forest checkpoints, DeSRA, and proposed method.

| Number of estimators | Max depth | | | | | |
|---|---|---|---|---|---|---|
| | 2 | 4 | 6 | 8 | 10 | 12 |
| 8 | 0.0250 | 0.0259 | 0.0304 | 0.0345 | 0.0362 | 0.0361 |
| 16 | 0.0250 | 0.0259 | 0.0304 | 0.0356 | 0.0366 | 0.0364 |
| 32 | 0.0250 | 0.0261 | 0.0299 | 0.0353 | 0.0365 | 0.0364 |
| DeSRA (t=0.3) | 0.0405 | | | | | |
| Proposed MLP method (t=0.15) | 0.0355 | | | | | |
| Proposed MLP method (t=0.3) | **0.0559** | | | | | |

the quality of artifact-prominence prediction. In particular, neither block averaging of ground-truth labels nor exclusion from the training of the regions that had not been labeled as containing artifacts during the subjective comparison.

### I.8. Filtering heatmaps through semantic segmentation

Similarly to Xie et al. (2023), we noticed that SR-artifact prominence is related to the affected object. Viewers hardly notice artifacts in objects such as grass, leaves, dirt, and soil. To confirm this hypothesis, we used the SAN (Xu et al., 2023) semantic segmentation method to cancel artifact predictions on pixels of certain semantic classes.

To select classes for exclusion, we analyzed the training set and calculated the average prominence for classes that occur in the artifact regions, then selected those classes with an average prominence less than 0.3. From these classes we manually removed those that define objects with potentially prominent artifacts—for example, "mouse," "umbrella," "cake," and so on. The remaining classes were used to exclude such objects from artifact prediction.

This procedure improves the $F1$-$score^{pr}$ metric (Table 21), which considers the artifact's prominence. Accordingly, excluding special classes with artifacts that are difficult to notice avoids penalizing the methods. However, this procedure fails to improve the results on the DeSRA dataset annotated in lab (Table 22), perhaps because the artifacts on organic matter were missed during annotation due to low prominence. Next, in a scenario with highly prominent test-set sampling, excluding classes only degrades IoU (Table 23, Table 24), because in this case all artifacts in the dataset are already prominent.

Considering these results, we have decided to omit this filtering step from our final proposed method, saving runtime and resources.

*Table 21.* SAN comparison on the full proposed dataset.

| | Original HR | | | | SPAN | | | | RLFN | | | |
|---|---|---|---|---|---|---|---|---|---|---|---|---|
| **Method** | $Prec^{pr}$ | $Rec^{pr}$ | **F1-score** | **PR-AUC** | $Prec^{pr}$ | $Rec^{pr}$ | **F1-score** | **PR-AUC** | $Prec^{pr}$ | $Rec^{pr}$ | **F1-score** | **PR-AUC** |
| LDL (t=0.005) | 0.0995 | 0.0159 | 0.0275 | 0.0039 | 0.0325 | 0.0141 | 0.0197 | 0.0034 | 0.0054 | 0.0026 | 0.0035 | 0.0007 |
| SSIM (t=0.55) | 0.0120 | 0.0169 | 0.0140 | 0.0022 | 0.0538 | 0.0269 | 0.0359 | 0.0083 | 0.0541 | 0.0252 | 0.0344 | 0.0054 |
| LPIPS (t=0.25) | 0.0680 | 0.0238 | 0.0352 | 0.0049 | 0.1094 | 0.0258 | 0.0418 | 0.0042 | 0.0893 | 0.0228 | 0.0364 | 0.0043 |
| ERQA (t=0.55) | 0.0024 | 0.0032 | 0.0028 | 0.0001 | -0.0163 | -0.0119 | -0.0137 | -0.0002 | -0.0262 | -0.0155 | -0.0195 | -0.0004 |
| PAL4Inpaint (bin., no-ref) | 0.0208 | 0.0081 | 0.0117 | N/A | 0.0208 | 0.0081 | 0.0117 | N/A | 0.0208 | 0.0081 | 0.0117 | N/A |
| PAL4VST (bin., no-ref) | 0.0464 | 0.0033 | 0.0062 | N/A | 0.0464 | 0.0033 | 0.0062 | N/A | 0.0464 | 0.0033 | 0.0062 | N/A |
| DISTS (t=0.25) | 0.0620 | 0.0503 | 0.0555 | 0.0062 | 0.1057 | 0.0530 | 0.0706 | 0.0085 | 0.1204 | 0.0499 | **0.0706** | 0.0082 |
| bd_jup (t=0.1) | 0.0032 | 0.0064 | 0.0043 | 0.0028 | 0.0088 | 0.0131 | 0.0105 | 0.0013 | 0.0063 | 0.0091 | 0.0074 | 0.0017 |
| ssm_jup (t=0.2) | 0.0266 | 0.0237 | 0.0251 | 0.0012 | 0.0165 | 0.0127 | 0.0144 | 0.0011 | 0.0209 | 0.0158 | 0.0180 | 0.0009 |
| DeSRA | 0.0781 | 0.0274 | 0.0405 | 0.0068 | 0.3159 | 0.0197 | 0.0371 | **0.0154** | 0.2861 | 0.0167 | 0.0315 | 0.0120 |
| **Ours (t=0.15)** | 0.0317 | 0.0402 | 0.0355 | 0.0121 | 0.0335 | 0.0315 | 0.0325 | 0.0075 | 0.0338 | 0.0290 | 0.0312 | 0.0075 |
| **Ours (t=0.3)** | 0.0762 | 0.0441 | 0.0559 | | 0.0503 | 0.0224 | 0.0310 | | 0.0581 | 0.0235 | 0.0334 | |
| LDL + SAN (t=0.005) | 0.1458 | 0.0171 | 0.0307 | 0.0055 | 0.1221 | 0.0305 | 0.0488 | 0.0104 | 0.1128 | 0.0271 | 0.0438 | 0.0081 |
| SSIM + SAN (t=0.55) | 0.0623 | 0.0532 | 0.0574 | 0.0098 | 0.1454 | 0.0421 | 0.0652 | 0.0142 | 0.1406 | 0.0384 | 0.0604 | 0.0116 |
| LPIPS + SAN (t=0.25) | 0.0766 | 0.0221 | 0.0344 | 0.0061 | 0.1560 | 0.0244 | 0.0421 | 0.0074 | 0.1416 | 0.0232 | 0.0399 | 0.0077 |
| ERQA + SAN (t=0.55) | 0.0579 | 0.0471 | 0.0520 | 0.0021 | 0.0559 | 0.0231 | 0.0327 | 0.0017 | 0.0568 | 0.0181 | 0.0275 | 0.0016 |
| PAL4Inpaint + SAN (bin., no-ref) | 0.0268 | 0.0091 | 0.0136 | N/A | 0.0268 | 0.0091 | 0.0136 | N/A | 0.0268 | 0.0091 | 0.0136 | N/A |
| PAL4VST + SAN (bin., no-ref) | 0.0419 | 0.0028 | 0.0053 | N/A | 0.0419 | 0.0028 | 0.0053 | N/A | 0.0419 | 0.0028 | 0.0053 | N/A |
| DISTS + SAN (t=0.25) | 0.0736 | 0.0479 | 0.0580 | 0.0088 | 0.1331 | 0.0493 | **0.0719** | 0.0090 | 0.1468 | 0.0460 | 0.0700 | 0.0098 |
| bd_jup + SAN (t=0.1) | 0.0393 | 0.0522 | 0.0448 | 0.0069 | 0.0571 | 0.0534 | 0.0552 | 0.0063 | 0.0552 | 0.0500 | 0.0525 | 0.0061 |
| ssm_jup + SAN (t=0.2) | 0.0810 | 0.0460 | 0.0587 | 0.0057 | 0.0824 | 0.0386 | 0.0526 | 0.0062 | 0.0867 | 0.0397 | 0.0544 | 0.0056 |
| DeSRA + SAN (t=0.3) | 0.0813 | 0.0245 | 0.0376 | 0.0072 | 0.3106 | 0.0176 | 0.0333 | 0.0150 | 0.2806 | 0.0147 | 0.0279 | 0.0118 |
| **Ours + SAN (t=0.15)** | 0.0796 | 0.0671 | 0.0728 | **0.0179** | 0.0944 | 0.0551 | 0.0696 | 0.0124 | 0.0991 | 0.0520 | 0.0682 | **0.0121** |
| **Ours + SAN (t=0.3)** | 0.1274 | 0.0542 | **0.0761** | | 0.1396 | 0.0380 | 0.0598 | | 0.1451 | 0.0369 | 0.0588 | |

*Table 22.* SAN comparison on the full DeSRA dataset.

| | MSE-SR | | | | SPAN | | | | RLFN | | | |
|---|---|---|---|---|---|---|---|---|---|---|---|---|
| **Method** | $Prec^{pr}$ | $Rec^{pr}$ | **F1-score** | **PR-AUC** | $Prec^{pr}$ | $Rec^{pr}$ | **F1-score** | **PR-AUC** | $Prec^{pr}$ | $Rec^{pr}$ | **F1-score** | **PR-AUC** |
| LDL (t=0.005) | 0.2320 | 0.1305 | 0.1670 | 0.0518 | 0.1615 | 0.1621 | **0.1618** | **0.0486** | 0.2339 | 0.1242 | 0.1622 | 0.0503 |
| SSIM (t=0.55) | 0.1929 | 0.1736 | 0.1828 | 0.0548 | 0.1211 | 0.1930 | 0.1488 | 0.0372 | 0.1819 | 0.1755 | 0.1786 | 0.0551 |
| LPIPS (t=0.25) | 0.1543 | 0.1268 | 0.1392 | 0.0349 | 0.1304 | 0.1445 | 0.1371 | 0.0300 | 0.1591 | 0.1351 | 0.1462 | 0.0389 |
| ERQA (t=0.55) | 0.0711 | 0.0275 | 0.0396 | 0.0026 | 0.0396 | 0.0590 | 0.0474 | 0.0017 | 0.0728 | 0.0275 | 0.0399 | 0.0025 |
| PAL4Inpaint (bin., no-ref) | 0.0543 | 0.0693 | 0.0609 | N/A | 0.0543 | 0.0693 | 0.0609 | N/A | 0.0543 | 0.0693 | 0.0609 | N/A |
| PAL4VST (bin., no-ref) | 0.0243 | 0.0030 | 0.0054 | N/A | 0.0243 | 0.0030 | 0.0054 | N/A | 0.0243 | 0.0030 | 0.0054 | N/A |
| DISTS (t=0.25) | 0.1478 | 0.1813 | 0.1628 | 0.0376 | 0.0717 | 0.2115 | 0.1071 | 0.0213 | 0.1470 | 0.1847 | 0.1637 | 0.0457 |
| bd_jup (t=0.1) | 0.0825 | 0.2043 | 0.1175 | 0.0230 | 0.0585 | 0.2153 | 0.0920 | 0.0135 | 0.0825 | 0.2079 | 0.1181 | 0.0244 |
| ssm_jup (t=0.2) | 0.1900 | 0.1655 | 0.1769 | 0.0377 | 0.1170 | 0.1825 | 0.1426 | 0.0250 | 0.1717 | 0.1663 | 0.1690 | 0.0346 |
| DeSRA | 0.2095 | 0.1505 | 0.1752 | 0.0579 | 0.1251 | 0.1298 | 0.1274 | 0.0273 | 0.1998 | 0.1473 | 0.1696 | 0.0550 |
| **Ours (t=0.15)** | 0.1565 | 0.2063 | 0.1780 | **0.0616** | 0.0849 | 0.2262 | 0.1235 | 0.0398 | 0.1500 | 0.2064 | 0.1737 | **0.0605** |
| **Ours (t=0.3)** | 0.2170 | 0.1701 | **0.1907** | | 0.1271 | 0.1955 | 0.1540 | | 0.2117 | 0.1727 | **0.1902** | |
| LDL + SAN (t=0.005) | 0.2483 | 0.1087 | 0.1512 | 0.0446 | 0.1786 | 0.1336 | 0.1528 | 0.0416 | 0.2534 | 0.1036 | 0.1471 | 0.0435 |
| SSIM + SAN (t=0.55) | 0.2238 | 0.1447 | 0.1758 | 0.0513 | 0.1386 | 0.1612 | 0.1491 | 0.0327 | 0.2119 | 0.1461 | 0.1729 | 0.0500 |
| LPIPS + SAN (t=0.25) | 0.1806 | 0.1035 | 0.1316 | 0.0328 | 0.1491 | 0.1211 | 0.1336 | 0.0259 | 0.1854 | 0.1100 | 0.1380 | 0.0342 |
| ERQA + SAN (t=0.55) | 0.0905 | 0.0254 | 0.0396 | 0.0030 | 0.0463 | 0.0506 | 0.0484 | 0.0018 | 0.0919 | 0.0257 | 0.0402 | 0.0029 |
| PAL4Inpaint + SAN (bin., no-ref) | 0.0556 | 0.0547 | 0.0552 | N/A | 0.0556 | 0.0547 | 0.0552 | N/A | 0.0556 | 0.0547 | 0.0552 | N/A |
| PAL4VST + SAN (bin., no-ref) | 0.0283 | 0.0028 | 0.0050 | N/A | 0.0283 | 0.0028 | 0.0050 | N/A | 0.0283 | 0.0028 | 0.0050 | N/A |
| DISTS + SAN (t=0.25) | 0.1748 | 0.1535 | 0.1634 | 0.0376 | 0.0820 | 0.1774 | 0.1121 | 0.0171 | 0.1742 | 0.1562 | 0.1647 | 0.0374 |
| bd_jup + SAN (t=0.1) | 0.0973 | 0.1737 | 0.1247 | 0.0210 | 0.0676 | 0.1825 | 0.0987 | 0.0124 | 0.0970 | 0.1763 | 0.1251 | 0.0219 |
| ssm_jup + SAN (t=0.2) | 0.2065 | 0.1393 | 0.1663 | 0.0321 | 0.1290 | 0.1522 | 0.1396 | 0.0216 | 0.1887 | 0.1399 | 0.1607 | 0.0297 |
| DeSRA + SAN (t=0.3) | 0.2460 | 0.1262 | 0.1668 | 0.0542 | 0.1435 | 0.1084 | 0.1235 | 0.0261 | 0.2340 | 0.1231 | 0.1614 | 0.0513 |
| **Ours + SAN (t=0.15)** | 0.1782 | 0.1739 | 0.1760 | 0.0563 | 0.0951 | 0.1887 | 0.1265 | 0.0367 | 0.1723 | 0.1743 | 0.1733 | 0.0549 |
| **Ours + SAN (t=0.3)** | 0.2384 | 0.1416 | 0.1777 | | 0.1390 | 0.1619 | 0.1496 | | 0.2324 | 0.1437 | 0.1776 | |

*Table 23.* SAN comparison on the prominent subset of the proposed dataset.

| Method | Original HR | | | | | SPAN | | | | | RLFN | | | | |
|---|---|---|---|---|---|---|---|---|---|---|---|---|---|---|---|
| | Precision | Recall | Prec * Rec | IoU | PR-AUC | Precision | Recall | Prec * Rec | IoU | PR-AUC | Precision | Recall | Prec * Rec | IoU | PR-AUC |
| LDL (t=0.005) | 0.3684 | 0.1418 | 0.0522 | 0.1043 | 0.1387 | 0.3764 | 0.3055 | 0.1150 | 0.1788 | 0.3110 | 0.3942 | 0.3091 | 0.1218 | 0.1779 | 0.3361 |
| SSIM (t=0.55) | 0.2917 | 0.6982 | 0.2036 | 0.3460 | 0.3802 | 0.4763 | 0.3455 | 0.1645 | 0.2642 | 0.3861 | 0.4752 | 0.3091 | 0.1469 | 0.2437 | 0.3710 |
| LPIPS (t=0.25) | 0.5876 | 0.2473 | 0.1453 | 0.2621 | 0.3860 | 0.4983 | 0.1418 | 0.0707 | 0.1340 | 0.3044 | 0.4393 | 0.1600 | 0.0703 | 0.1324 | 0.2971 |
| ERQA (t=0.55) | 0.1957 | 0.6218 | 0.1217 | 0.2495 | 0.1220 | 0.2300 | 0.3236 | 0.0744 | 0.1670 | 0.1052 | 0.2494 | 0.2327 | 0.0581 | 0.1492 | 0.1052 |
| PAL4Inpaint (bin., no-ref) | 0.1833 | 0.1164 | 0.0213 | 0.0753 | N/A | 0.1833 | 0.1164 | 0.0213 | 0.0753 | N/A | 0.1833 | 0.1164 | 0.0213 | 0.0753 | N/A |
| PAL4VST (bin., no-ref) | 0.2682 | 0.0327 | 0.0088 | 0.0463 | N/A | 0.2682 | 0.0327 | 0.0088 | 0.0463 | N/A | 0.2682 | 0.0327 | 0.0088 | 0.0463 | N/A |
| DISTS (t=0.25) | 0.2947 | 0.6182 | 0.1822 | 0.3525 | 0.2620 | 0.4572 | 0.4182 | 0.1912 | 0.2820 | 0.3242 | 0.4726 | 0.3818 | 0.1804 | 0.2783 | 0.3386 |
| bd_jup (t=0.1) | 0.1810 | 0.8364 | 0.1514 | 0.2843 | 0.3311 | 0.1800 | 0.6945 | 0.1250 | 0.2475 | 0.2342 | 0.1983 | 0.6509 | 0.1291 | 0.2434 | 0.2275 |
| ssm_jup (t=0.2) | 0.2350 | 0.4909 | 0.1153 | 0.2368 | 0.2127 | 0.2383 | 0.4255 | 0.1014 | 0.2133 | 0.2273 | 0.2689 | 0.4582 | 0.1232 | 0.2221 | 0.2411 |
| DeSRA | 0.4791 | 0.2764 | 0.1324 | 0.2560 | 0.3173 | 0.6976 | 0.0982 | 0.0685 | 0.1296 | 0.3358 | 0.7040 | 0.0836 | 0.0589 | 0.1205 | 0.3025 |
| **Ours (t=0.15)** | 0.2447 | 0.8145 | 0.1993 | 0.3639 | 0.4756 | 0.3709 | 0.5636 | **0.2090** | **0.3018** | 0.3931 | 0.3890 | 0.5273 | **0.2051** | **0.2903** | 0.3829 |
| **Ours (t=0.3)** | 0.4357 | 0.5745 | **0.2503** | **0.3669** | | 0.5209 | 0.3345 | 0.1743 | 0.2357 | | 0.5138 | 0.3418 | 0.1756 | 0.2311 | |
| LDL + SAN (t=0.005) | 0.3927 | 0.1236 | 0.0486 | 0.0964 | 0.1296 | 0.4220 | 0.2764 | 0.1166 | 0.1692 | 0.3086 | 0.4260 | 0.2800 | 0.1193 | 0.1680 | 0.3179 |
| SSIM + SAN (t=0.55) | 0.3164 | 0.6436 | 0.2036 | 0.3476 | 0.3759 | 0.5175 | 0.3164 | 0.1637 | 0.2500 | 0.3608 | 0.5315 | 0.2800 | 0.1488 | 0.2294 | 0.3552 |
| LPIPS + SAN (t=0.25) | 0.6108 | 0.2145 | 0.1310 | 0.2417 | 0.3694 | 0.5437 | 0.1273 | 0.0692 | 0.1224 | 0.2750 | 0.4918 | 0.1382 | 0.0680 | 0.1189 | 0.2748 |
| ERQA + SAN (t=0.55) | 0.1979 | 0.5927 | 0.1173 | 0.2459 | 0.1194 | 0.2497 | 0.3091 | 0.0772 | 0.1605 | 0.1067 | 0.2745 | 0.2182 | 0.0599 | 0.1428 | 0.1055 |
| PAL4Inpaint + SAN (bin., no-ref) | 0.1780 | 0.1127 | 0.0201 | 0.0706 | N/A | 0.1780 | 0.1127 | 0.0201 | 0.0706 | N/A | 0.1780 | 0.1127 | 0.0201 | 0.0706 | N/A |
| PAL4VST + SAN (bin., no-ref) | 0.2591 | 0.0291 | 0.0075 | 0.0402 | N/A | 0.2591 | 0.0291 | 0.0075 | 0.0402 | N/A | 0.2591 | 0.0291 | 0.0075 | 0.0402 | N/A |
| DISTS + SAN (t=0.25) | 0.3167 | 0.5600 | 0.1773 | 0.3339 | 0.3491 | 0.4842 | 0.3673 | 0.1778 | 0.2570 | 0.3217 | 0.5069 | 0.3345 | 0.1696 | 0.2525 | 0.3260 |
| bd_jup + SAN (t=0.1) | 0.1722 | 0.7600 | 0.1309 | 0.2744 | 0.3243 | 0.1821 | 0.6218 | 0.1132 | 0.2360 | 0.2226 | 0.1990 | 0.5818 | 0.1158 | 0.2329 | 0.2112 |
| ssm_jup + SAN (t=0.2) | 0.2490 | 0.4473 | 0.1114 | 0.2247 | 0.2026 | 0.2454 | 0.3855 | 0.0946 | 0.2007 | 0.2072 | 0.2830 | 0.4145 | 0.1173 | 0.2101 | 0.2319 |
| DeSRA + SAN (t=0.3) | 0.5160 | 0.2545 | 0.1313 | 0.2419 | 0.3041 | 0.7193 | 0.0909 | 0.0654 | 0.1225 | 0.3120 | 0.7069 | 0.0764 | 0.0540 | 0.1128 | 0.2763 |
| **Ours + SAN (t=0.15)** | 0.2588 | 0.7455 | 0.1929 | 0.3476 | 0.4464 | 0.3897 | 0.5091 | 0.1984 | 0.2853 | 0.3765 | 0.4037 | 0.4764 | 0.1923 | 0.2735 | 0.3690 |
| **Ours + SAN (t=0.3)** | 0.4620 | 0.5236 | 0.2419 | 0.3410 | | 0.5587 | 0.2909 | 0.1625 | 0.2176 | | 0.5415 | 0.3091 | 0.1674 | 0.2127 | |

*Table 24.* SAN comparison on the prominent subset of the DeSRA dataset.

| Method | MSE-SR | | | | | SPAN | | | | | RLFN | | | | |
|---|---|---|---|---|---|---|---|---|---|---|---|---|---|---|---|
| | Precision | Recall | Prec * Rec | IoU | PR-AUC | Precision | Recall | Prec * Rec | IoU | PR-AUC | Precision | Recall | Prec * Rec | IoU | PR-AUC |
| LDL (t=0.005) | 0.5929 | 0.3772 | 0.2237 | 0.3724 | 0.4687 | 0.2977 | 0.5409 | 0.1610 | 0.3896 | 0.3418 | 0.5744 | 0.3467 | 0.1992 | 0.3443 | 0.4183 |
| SSIM (t=0.55) | 0.5338 | 0.6581 | 0.3513 | 0.5327 | 0.5730 | 0.2685 | 0.7560 | 0.2030 | **0.4590** | 0.3861 | 0.4977 | 0.6677 | 0.3323 | 0.5243 | **0.6051** |
| LPIPS (t=0.25) | 0.5152 | 0.4398 | 0.2266 | 0.3759 | 0.4488 | 0.3385 | 0.4526 | 0.1532 | 0.3450 | 0.3193 | 0.5261 | 0.5072 | 0.2669 | 0.4094 | 0.5014 |
| ERQA (t=0.55) | 0.1365 | 0.0401 | 0.0055 | 0.0523 | 0.0100 | 0.0618 | 0.2295 | 0.0142 | 0.0684 | 0.0154 | 0.1422 | 0.0337 | 0.0048 | 0.0514 | 0.0087 |
| PAL4Inpaint (bin., no-ref) | 0.1184 | 0.1958 | 0.0232 | 0.1139 | N/A | 0.1184 | 0.1958 | 0.0232 | 0.1139 | N/A | 0.1184 | 0.1958 | 0.0232 | 0.1139 | N/A |
| PAL4VST (bin., no-ref) | 0.0407 | 0.0177 | 0.0007 | 0.0140 | N/A | 0.0407 | 0.0177 | 0.0007 | 0.0140 | N/A | 0.0407 | 0.0177 | 0.0007 | 0.0140 | N/A |
| DISTS (t=0.25) | 0.3898 | 0.7400 | 0.2884 | 0.4919 | 0.4408 | 0.1097 | 0.8604 | 0.0944 | 0.3479 | 0.2290 | 0.3801 | 0.7576 | 0.2880 | 0.5016 | 0.5428 |
| bd_jup (t=0.1) | 0.1245 | 0.8347 | 0.1039 | 0.3580 | 0.1609 | 0.0919 | 0.8652 | 0.0795 | 0.2798 | 0.1221 | 0.1160 | 0.8555 | 0.0992 | 0.3625 | 0.1773 |
| ssm_jup (t=0.2) | 0.4629 | 0.5120 | 0.2370 | 0.4032 | 0.3889 | 0.3018 | 0.6164 | 0.1860 | 0.3770 | 0.2736 | 0.4087 | 0.5185 | 0.2119 | 0.3930 | 0.3646 |
| DeSRA | 0.6794 | 0.6228 | **0.4231** | 0.5277 | **0.6928** | 0.3324 | 0.4462 | 0.1483 | 0.3707 | 0.2910 | 0.6366 | 0.6100 | **0.3883** | 0.5082 | 0.6614 |
| **Ours (t=0.15)** | 0.4121 | 0.7961 | 0.3281 | **0.5420** | 0.6104 | 0.1633 | 0.8973 | 0.1465 | 0.4010 | 0.3873 | 0.3353 | 0.8427 | 0.2825 | **0.5301** | 0.6030 |
| **Ours (t=0.3)** | 0.6088 | 0.5618 | 0.3420 | 0.4866 | | 0.2970 | 0.7175 | **0.2131** | 0.4374 | | 0.5543 | 0.6276 | 0.3479 | 0.5049 | |
| LDL + SAN (t=0.005) | 0.5885 | 0.3114 | 0.1833 | 0.3007 | 0.3662 | 0.3039 | 0.4141 | 0.1259 | 0.3047 | 0.2717 | 0.5809 | 0.2921 | 0.1697 | 0.2814 | 0.3322 |
| SSIM + SAN (t=0.55) | 0.5145 | 0.5088 | 0.2618 | 0.4092 | 0.4605 | 0.2510 | 0.5955 | 0.1495 | 0.3510 | 0.2765 | 0.4908 | 0.5169 | 0.2537 | 0.4027 | 0.4519 |
| LPIPS + SAN (t=0.25) | 0.4990 | 0.3162 | 0.1578 | 0.2754 | 0.3175 | 0.3184 | 0.3483 | 0.1109 | 0.2588 | 0.2153 | 0.5128 | 0.3708 | 0.1901 | 0.3014 | 0.3433 |
| ERQA + SAN (t=0.55) | 0.1534 | 0.0401 | 0.0062 | 0.0481 | 0.0117 | 0.0672 | 0.1990 | 0.0134 | 0.0627 | 0.0134 | 0.1664 | 0.0337 | 0.0056 | 0.0475 | 0.0100 |
| PAL4Inpaint + SAN (bin., no-ref) | 0.1150 | 0.1621 | 0.0186 | 0.0859 | N/A | 0.1150 | 0.1621 | 0.0186 | 0.0859 | N/A | 0.1150 | 0.1621 | 0.0186 | 0.0859 | N/A |
| PAL4VST + SAN (bin., no-ref) | 0.0448 | 0.0177 | 0.0008 | 0.0130 | N/A | 0.0448 | 0.0177 | 0.0008 | 0.0130 | N/A | 0.0448 | 0.0177 | 0.0008 | 0.0130 | N/A |
| DISTS + SAN (t=0.25) | 0.3723 | 0.5698 | 0.2121 | 0.3733 | 0.3134 | 0.1052 | 0.6822 | 0.0718 | 0.2653 | 0.1363 | 0.3658 | 0.5811 | 0.2125 | 0.3824 | 0.3219 |
| bd_jup + SAN (t=0.1) | 0.1217 | 0.6613 | 0.0805 | 0.2887 | 0.1253 | 0.0823 | 0.6918 | 0.0569 | 0.2256 | 0.0872 | 0.1118 | 0.6822 | 0.0763 | 0.2936 | 0.1243 |
| ssm_jup + SAN (t=0.2) | 0.4549 | 0.4125 | 0.1876 | 0.3294 | 0.3165 | 0.2835 | 0.4848 | 0.1374 | 0.3020 | 0.2158 | 0.4032 | 0.4173 | 0.1683 | 0.3252 | 0.2963 |
| DeSRA + SAN (t=0.3) | 0.6685 | 0.4880 | 0.3262 | 0.4059 | 0.5389 | 0.3211 | 0.3451 | 0.1108 | 0.2779 | 0.2211 | 0.6269 | 0.4751 | 0.2978 | 0.3902 | 0.5166 |
| **Ours + SAN (t=0.15)** | 0.3955 | 0.6292 | 0.2489 | 0.4222 | 0.4864 | 0.1496 | 0.7143 | 0.1068 | 0.3080 | 0.2981 | 0.3646 | 0.6372 | 0.2323 | 0.4197 | 0.4832 |
| **Ours + SAN (t=0.3)** | 0.5810 | 0.4398 | 0.2555 | 0.3758 | | 0.2773 | 0.5602 | 0.1554 | 0.3360 | | 0.5723 | 0.4655 | 0.2664 | 0.3851 | |

