# OpenReview forum: "Prominence-Aware Artifact Detection and Dataset for Image Super-Resolution"
_ICML.cc/2026/Conference — Submitted to ICML 2026_

### Official Review · Reviewer_Rf4H · 2026-03-03

**Soundness:** 3
**Presentation:** 2
**Significance:** 2
**Originality:** 3
**Overall Recommendation:** 3
**Confidence:** 4

**Summary:**

This paper addresses the artifact detection problem in image super-resolution (SR) by proposing a novel perspective: characterizing artifacts based on their prominence to human observers, rather than traditional binary classification. The authors constructed a dataset containing 1302 artifact samples, each with a crowdsourced saliency score, and supplemented it with saliency annotations for 593 artifacts from the existing DeSRA dataset. Based on this dataset, they trained a lightweight regressor to generate a spatial saliency heatmap. Experiments show that this proposed method outperforms existing methods in artifact detection and can be used to guide fine-tuning of SR models to suppress artifacts.

**Compliance With Llm Reviewing Policy:**

Affirmed.

**Final Justification:**

After reviewing the authors' rebuttal, I decided to maintain my original weak-reject rating.

While the rebuttal alleviated some of my concerns about hyperparameter sensitivity and ablation experiments, the core issues remain. The authors' response to W2/Q1 revealed a **new flaw** rather than resolving the original one: the results of the continuous significance weighted fine-tuning experiment were slightly inferior to the binary masking method. This directly weakens the paper's core argument that continuous significance modeling has practical advantages over binary detection. Furthermore, the issue of data leakage remains poorly addressed.

**Key Questions For Authors:**

(1) Examining the Incremental Value of Saliency Continuous Modeling
In the core application of SR model fine-tuning, could a set of control experiments be added to directly compare the differences between "using saliency-weighted masks" and "using ordinary binary masks with the same detection method" on several metrics? I think this would directly answer the core question: "How much practical benefit does saliency modeling actually bring compared to binary detection?"

(2) Considering the Fundamental Reasons for MLP's Superiority to CNN
The CNN architecture in the appendix failed to surpass pixel-wise MLP. What is the fundamental reason for this counterintuitive result? The authors believe it's because "the input features have encoded sufficient context," but can this hypothesis be verified experimentally—for example, if local features without contextual information are input into the CNN, will its performance further decline? Or, is the size of 374 training samples limiting the performance of the CNN?

(3) Sensitivity of the evaluation metric κ
How much does the selection of the κ=0.3 threshold affect the stability of the method ranking? If κ is set to 0.2 and 0.4 respectively, will the relative rankings of the methods in Tables 2 and 3 change significantly? If the ranking is sensitive to κ, the reliability of the evaluation conclusions will be questioned; if the ranking is robust, it is recommended to explicitly demonstrate this robustness in the main text.

**Limitations:**

yes

**Strengths And Weaknesses:**

Strengths

(1) The core observation presented in the paper is valuable—48% of artifacts in the existing DeSRA dataset have a significance level below 50%, revealing the limitations of the binary annotation scheme. Re-examining artifact detection from a human perception perspective is a reasonable research direction.

(2) The dataset's contribution is substantial: the authors conducted extensive manual annotation work, including crowdsourced saliency scoring, providing new annotation resources for the community. The dataset construction method is relatively systematic, covering 11 SR methods, and is representative.

(3) The experimental design is comprehensive: the paper evaluated multiple datasets, including comparative experiments with existing methods, and demonstrated its application value in model fine-tuning.

(4) The writing is clear: the paper's overall expression is clear, the problem definition is explicit, and the review of relevant work is relatively thorough.

Weaknesses

(1) Lack of in-depth analysis and explanation of the superiority of MLP over CNN architecture
The authors have experimented with CNN (5-layer residual blocks) and various random forest configurations in the appendix, all of which failed to surpass the simple MLP. This result itself deserves further investigation. However, the paper only briefly mentions it as a basis for architecture selection without deeply analyzing the underlying reasons: Is it that the input features (DISTS, bd_jup, ssm_jup) have already encoded enough contextual information to make the complexity of the fusion architecture irrelevant, or is the scale of the training data (374 samples) itself a bottleneck for more complex models? In-depth analysis of this issue will provide important insights into understanding the essence of the saliency prediction problem.

(2) Insufficient quantification of the gain of saliency modeling relative to binary detection
The core motivation of the paper is that "binary labeling is insufficient to reflect the perceptual impact of artifacts," but the authors have not designed direct comparative experiments to quantify the actual value of this gap. Specifically, in the core downstream task of SR model fine-tuning, how much incremental benefit does the mask constructed using continuous saliency scores bring compared to the ordinary binary mask? Table 6 only compares different detection methods, without controlling for the variable "whether significance weighting is used", making it difficult to assess the contribution of significance modeling itself.

(3) Ineffective Decoupling of Dataset and Method Contributions
The performance improvement in this paper may come from two independent sources: (a) the new dataset provides richer and more accurate training signals; (b) the three-feature fusion method itself is superior. However, the paper lacks crucial experiments to decouple these two aspects. For example, how much would the performance improve if existing methods such as DeSRA were retrained on the new dataset? The lack of this comparative experiment makes it difficult to clarify the independent contributions of the paper's method itself.

(4) Lack of Sensitivity Analysis for Key Hyperparameter κ in Evaluation Metrics
The significance-weighted precision/recall metric proposed in this paper introduces a boundary value κ=0.3. The authors describe this setting as "guided by dataset statistics and observer consistency," but do not provide a systematic sensitivity analysis. Will different κ values ​​change the ranking order between methods? The choice of this hyperparameter is crucial to the effectiveness of the entire evaluation system, and its rationale needs more thorough justification.

(5) I am concerned about the potential data leakage risk in the fine-tuning experiment.
The SR fine-tuning experiment in Table 6 uses the DeSRA dataset for both training and testing (split by the target SR model), and the binarization threshold for each method is selected by maximizing the Precision × Recall product on the DeSRA saliency subset (Section 5.2). This setting may cause implicit data leakage, have an uneven impact on different methods, and thus affect the fairness of the fine-tuning experiment conclusions.

---

> ### Author Rebuttal · Authors · 2026-03-31
>
> Thank you for your thorough review and constructive feedback! We’d like to address the weaknesses and questions.
>
> > (2) Insufficient quantification of the gain of saliency modeling relative to binary detection
>
> > (1) Examining the Incremental Value of Saliency Continuous Modeling In the core application of SR model fine-tuning, could a set of control experiments be added to directly compare the differences between "using saliency-weighted masks" and "using ordinary binary masks with the same detection method" on several metrics? I think this would directly answer the core question: "How much practical benefit does saliency modeling actually bring compared to binary detection?"
>
> > (5) I am concerned about the potential data leakage risk in the fine-tuning experiment. The SR fine-tuning experiment in Table 6 uses the DeSRA dataset for both training and testing (split by the target SR model), and the binarization threshold for each method is selected by maximizing the Precision × Recall product on the DeSRA saliency subset (Section 5.2).
>
> For the SR fine-tuning experiment, we used a binary mask so that we could construct artificial GT training samples by replacing the pixels within that mask with artifact-free RLFN upscaling. We also tried a continuous adaptation of this fine-tuning experiment, where we blended between artifact-free and original pixels, weighted by the predicted prominence. This experiment yielded similar but slightly worse results for all artifact detection methods in the end, likely due to confusing training signals for the SR model. So prominence modeling doesn’t *directly* improve the fine-tuning experiment. Since no thresholds were used in this continuous version of the experiment, it should also address the data leakage concern.
>
> We believe that prominence modeling helps the most a step earlier. It allows to train a better artifact detection method that ultimately provides a better signal for downstream tasks such as SR fine-tuning.
>
> > (2) Considering the Fundamental Reasons for MLP's Superiority to CNN
> >
> > Or, is the size of 374 training samples limiting the performance of the CNN?
>
> We believe the small size of our dataset is indeed the limiting factor for a CNN. Training a CNN with just RGB inputs (no high-level features) produces unsatisfactory performance. Training a CNN on the same input features as mentioned in Appendix I.6 yields similar results to the MLP.
>
> > (3) Ineffective Decoupling of Dataset and Method Contributions
> >
> > For example, how much would the performance improve if existing methods such as DeSRA were retrained on the new dataset?
>
> LDL and DeSRA, the two methods in our comparison that specifically address the artifact detection problem, are both not learning-based, so they unfortunately cannot be retrained on our dataset.
>
> However, we did an alternative experiment: we trained our own method on the DeSRA dataset, and then evaluated it on ours. Here are the results:
>
> ||Original HR||SPAN||RLFN||
> |---|---|---|---|---|---|---|
> |**Method**|**F1-score**|**PR-AUC**|**F1-score**|**PR-AUC**|**F1-score**|**PR-AUC**|
> |DISTS|*0.0555*|0.0062|**0.0706**|*0.0085*|**0.0706**|*0.0082*|
> |DeSRA|0.0405|*0.0068*|*0.0371*|**0.0154**|0.0315|**0.0120**|
> |Ours (t=0.3)|**0.0559**|**0.0121**|0.0310|0.0075|0.0334|0.0075|
> |Ours-DeSRA (t=0.3)|0.0341|0.0036|0.0367|0.0036|*0.0337*|0.0023|
>
> And on the prominent subset:
>
> ||Original HR||SPAN||RLFN||
> |---|---|---|---|---|---|---|
> |**Method**|**IoU**|**PR-AUC**|**IoU**|**PR-AUC**|**IoU**|**PR-AUC**|
> |DISTS|*0.3525*|0.2619|*0.2820*|0.3231|*0.2783*|*0.3386*|
> |DeSRA|0.2560|0.3173|0.1296|*0.3357*|0.1205|0.3025|
> |Ours (t=0.3)|**0.3669**|**0.4756**|0.2355|**0.3931**|0.2311|**0.3829**|
> |Ours-DeSRA (t=0.3)|0.3155|*0.3372*|**0.2825**|0.3017|**0.2793**|0.3046|
>
> As you can see, on our full dataset the results range from considerably worse to ever so slightly better, while on the prominent subset they are consistently competitive. This shows that both our dataset and method contribute to the performance: our dataset helps our method accurately model less prominent artifacts (training without it degrades performance), whereas our method is by itself competitive on prominent artifacts (even when training without our dataset).
>
> > (3) Sensitivity of the evaluation metric κ
>
> We followed your suggestion and measured our F1-scoring with κ set to 0.2 and 0.4, then computed the Avg. Ranks by PR-AUC (same as Table 2). κ=0.3 matches Table 2 in our paper.
>
> |Method|κ=0.2|κ=0.3|κ=0.4|
> |---|---|---|---|
> |LDL|4.0|4.7|3.3|
> |SSIM|3.7|3.7|4.0|
> |LPIPS|5.0|5.2|5.8|
> |ERQA|9.8|9.8|10.0|
> |PaQ-2-PiQ|11.0|10.2|8.7|
> |TOPIQ|9.2|10.0|9.2|
> |DISTS|4.5|4.2|4.3|
> |bd\_jup|7.3|7.2|9.0|
> |ssm\_jup|7.2|6.8|6.8|
> |DeSRA|*2.7*|*2.3*|*2.7*|
> |Ours|**1.7**|**2.0**|**2.2**|
>
> As you can see, our method remains in the first place across all three κ values. The relative rankings of other methods also remain mostly stable. Table 3 is using the DeSRA scoring, so it is unaffected by our κ.

---

> > ### Author Rebuttal · Reviewer_Rf4H · 2026-04-01
> >
> > The author provided a complete response to my questions, but I noticed that some of the answers did not reassure me enough to raise my score.
> > In the first response, the author acknowledged a newly discovered flaw: **"prominence modeling doesn't directly improve the fine-tuning experiment."** This also led to my continued concern about the risk of data leakage, as the author did not seem to explain in depth whether the risk of leakage in the original binary experiment truly existed and its extent.
> >
> > Therefore, I plan to maintain my original score.

---

> > > ### Author Response · Authors · 2026-04-08
> > >
> > > Thank you for your acknowledgement. We’ve now had more time and were able to rerun some of the fine-tuning experiments.
> > >
> > > We mentioned that threshold-free prominence-weighted blending for constructing artificial GT yielded similar results, which we believe is due to confusing training signal, since pixels with artifacts end up partially blended with clean pixels.
> > >
> > > We’ve now run a second experiment that does not have this shortcoming. Here, metric outputs are still binarized with thresholds to create artificial GT, but the loss contribution of the resulting training sample is weighted by the prominence, computed as the mean predicted heatmap value inside the binarized artifact region. This way, fine-tuning has a stronger focus on more prominent artifacts.
> > >
> > > |Target SR:|LDL|||||\||RealESRGAN|||||\||SwinIR|||||
> > > |---|---|---|---|---|---|---|---|---|---|---|---|---|---|---|---|---|---|
> > > |**Method**|$\Delta IoU^↑$|${Add_{img}}^↓$|${Rem_{img}}^↑$|${Add_{pix}}^↓$|${Rem_{pix}}^↑$|\||$\Delta IoU^↑$|${Add_{img}}^↓$|${Rem_{img}}^↑$|${Add_{pix}}^↓$|${Rem_{pix}}^↑$|\||$\Delta IoU^↑$|${Add_{img}}^↓$|${Rem_{img}}^↑$|${Add_{pix}}^↓$|${Rem_{pix}}^↑$|
> > > |DISTS + w. loss|27.75|24.47|19.15|**0.01**|95.44|\||33.38|40.20|20.60|0.23|97.47|\||15.04|34.31|14.22|0.02|93.26|
> > > |Ours + w. loss|**35.25**|**15.43**|**55.32**|0.02|**98.24**|\||**38.15**|**8.04**|**68.34**|**0.01**|**98.44**|\||11.99|11.27|**57.84**|**0.00**|91.94|
> > > |DISTS|27.00|33.51|11.70|0.03|93.27|\||33.47|32.66|15.58|0.03|96.49|\||**15.23**|26.47|13.73|0.02|**93.38**|
> > > |Ours|34.71|20.74|45.21|0.03|97.20|\||38.01|14.57|55.78|0.03|98.09|\||11.86|**6.37**|57.35|**0.00**|91.95|
> > >
> > > These results show that taking predicted prominence into account *does* benefit fine-tuning. Note that our method, which we explicitly train to model prominence, benefits almost everywhere from this setting, whereas DISTS shows inconsistent gains.
> > >
> > > We’d also like to reemphasize: we believe prominence modeling helps to train a better detector; this benefit then flows to all downstream tasks, even if those don’t directly use the prominence output.
> > >
> > > To address the threshold selection data leakage concern, we reran fine-tuning with slightly higher and lower thresholds:
> > >
> > > |Target SR:|LDL|||||\||RealESRGAN|||||\||SwinIR|||||
> > > |---|---|---|---|---|---|---|---|---|---|---|---|---|---|---|---|---|---|
> > > |**Method**|$\Delta IoU^↑$|${Add_{img}}^↓$|${Rem_{img}}^↑$|${Add_{pix}}^↓$|${Rem_{pix}}^↑$|\||$\Delta IoU^↑$|${Add_{img}}^↓$|${Rem_{img}}^↑$|${Add_{pix}}^↓$|${Rem_{pix}}^↑$|\||$\Delta IoU^↑$|${Add_{img}}^↓$|${Rem_{img}}^↑$|${Add_{pix}}^↓$|${Rem_{pix}}^↑$|
> > > |DISTS (t=0.3)|23.14|32.45|10.64|**0.03**|80.38|\||27.45|22.11|22.11|0.23|91.10|\||**10.43**|7.35|23.04|0.00|**84.28**|
> > > |DeSRA (t=0.35)|27.21|48.94|29.79|0.55|70.63|\||30.60|30.15|**58.79**|0.06|80.03|\||8.51|23.04|32.84|0.06|50.63|
> > > |Ours (t=0.4)|**33.19**|**20.21**|**32.98**|0.06|**93.77**|\||**33.59**|**18.59**|39.20|**0.01**|**95.18**|\||9.47|**2.45**|**51.47**|**0.00**|83.37|
> > > |---||||||||||||||||||
> > > |DISTS (t=0.25)|27.00|33.51|11.70|**0.03**|93.27|\||33.47|32.66|15.58|**0.03**|96.49|\||**15.23**|26.47|13.73|0.02|**93.38**|
> > > |DeSRA (t=0.3)|29.18|54.26|25.00|0.58|71.02|\||33.66|34.17|**56.78**|0.25|80.06|\||8.98|32.84|28.43|0.12|51.69|
> > > |Ours (t=0.3)|**34.71**|**20.74**|**45.21**|**0.03**|**97.20**|\||**38.01**|**14.57**|55.78|**0.03**|**98.09**|\||11.86|**6.37**|**57.35**|**0.00**|91.95|
> > > |---||||||||||||||||||
> > > |DISTS (t=0.2)|19.47|64.36|4.79|0.09|94.39|\||24.39|63.82|6.03|0.09|95.40|\||11.67|61.27|3.43|0.08|91.31|
> > > |DeSRA (t=0.25)|32.02|54.26|26.60|0.66|75.13|\||35.97|39.20|51.26|0.17|82.10|\||10.07|31.86|37.25|0.12|60.97|
> > > |Ours (t=0.25)|**34.60**|**15.96**|**54.26**|**0.03**|**98.48**|\||**39.04**|**20.10**|**55.78**|**0.03**|**98.92**|\||**14.40**|**17.16**|**50.00**|**0.01**|**95.08**|
> > >
> > > As these results show, our method remains competitive in all settings, even when the thresholds aren’t (indirectly) derived from the DeSRA dataset.
> > >
> > > We believe the above clarifications address the remaining concerns and clarify the performance characteristics of our method. We hope they resolve the follow-up questions and support a positive reassessment of our work.

---

### Official Review · Reviewer_f3QX · 2026-03-05

**Soundness:** 3
**Presentation:** 3
**Significance:** 3
**Originality:** 2
**Overall Recommendation:** 4
**Confidence:** 3

**Summary:**

This paper presents a prominence-aware artifact detection framework and a specialized dataset of artifacts produced by 11 contemporary SISR methods. The authors identify that existing detection methods fail to distinguish between barely noticeable defects and highly disturbing ones by treating them as uniform binary masks.
To solve this, they introduce three main components:
• Crowdsourced Prominence Dataset: A collection of 1,302 artifact examples from 11 contemporary SISR methods, each annotated with a prominence score derived from 30 separate human assessments. This approach reveals that nearly half of existing benchmark artifacts go unnoticed by viewers, highlighting the need for perceptual weighting.
• Artifacts detector with Lightweight Fusion Module: A 3-layer multilayer perceptron (MLP) regressor that aggregates features from existing quality metrics (DISTS, ssm_jup, and bd_jup) to produce spatial prominence heatmaps . This module captures both fine-grained texture distortions and broader perceptual quality to quantify artifact severity.
• Pseudo-GT Evaluation Pipeline: A practical framework that adapts full-reference metrics for real-world scenarios where high-resolution ground truth is unavailable. By utilizing a lightweight, artifact-robust off-the-shelf SR model (RLFN) as a pseudo ground truth, the method maintains high detection accuracy without requiring original HR frames .
The notable contribution is the development of a metric that not only outperforms existing detectors in aligning with human perception but also effectively guides the fine-tuning of SR models to suppress visually disturbing artifacts .

**Compliance With Llm Reviewing Policy:**

Affirmed.

**Final Justification:**

**Initial Assessment & Strengths/Weaknesses**: This paper tackles the problem of Single Image Super-Resolution (SISR) artifact detection by introducing a pseudo-GT evaluation pipeline and a crowd sourced prominence dataset. However, my initial assessment raised several concerns regarding dataset bias, feature redundancy, statistical claims on post-processing, and dependency on pseudo-GT fidelity.

**Rebuttal Evaluation**: During the rebuttal, the authors addressed my several concerns. They provided reasonable justifications by clarifying that the low-prominence samples serve as necessary negative signals to prevent the model from over-fitting to imperceptible noise. Additionally, their experimental design choice of using NN-upscaled images as a reference effectively prevents the model from penalizing plausible high-frequency hallucinations. The authors also committed to five key revisions for the camera-ready version:

(1) Recalibrating post-processing claims: Reframing the mask post-processing as a visual formatting step rather than a direct performance improvement.

(2) Justifying feature fusion with quantitative and visual evidence: Including a table of FLOPs, parameters, and inference time, supplemented by visual ablation examples (e.g., DISTS-only vs. all features) to explicitly demonstrate the claimed benefits of the multi-feature architecture on specific textures.

(3) Including a DeSRA dataset comparison: Adding experimental results to prove that prominence-based annotations provide a richer training signal, particularly for complex distortions that binary-mask benchmarks often fail to capture.

(4) discussing pseudo-GT limitations: Incorporating a main-text discussion on potential failure cases and mitigation strategies when reliable references are unavailable.

(5) documenting the "Original" image methodology: Explicitly detailing the use of NN-upscaled images in the annotation process to ensure reproducibility and clarify the evaluation criteria.

**Final Recommendation**: The authors' clarifications and their commitments to revise the manuscript have mitigated my methodological concerns. Assuming the promised changes are fully implemented, the paper meets the threshold for acceptance. Therefore, I maintain my **Weak Accept** rating.

**Key Questions For Authors:**

Questions about weaknesses:
   1. Statistical Significance of Post-processing: In Section 3.2, you state that mask-postprocessing improved mean artifact prominence from 47.7% to 49.4%. However, Appendix B reports that with 30 assessors, the 95% confidence interval is approximately ±20%. How do you justify the effectiveness of this algorithm when the claimed improvement (1.7%p) is significantly smaller than the statistical margin of error for your viewer count?
   2. Mitigating Dataset Bias: Section G and Section 3.3 acknowledge a bias toward low-prominence samples, with nearly half of the masks falling below 0.3 prominence. How did you ensure that the MLP regressor learned to identify truly disturbing, high-prominence artifacts rather than simply overfitting to low-level texture noises that are barely visible to humans?
   3. Pseudo-GT Reliability in High-Frequency Regions: You mention that false detections can occur when the lightweight SR (RLFN) fails to reconstruct fine textures. In real-world applications where HR references are unavailable, how does the framework distinguish between a target SR's actual artifact and a pseudo-GT's reconstruction failure? Is there a risk of penalizing high-quality SR models that restore textures better than the RLFN reference?
   4. Conceptual Overlap and Feature Redundancy: DISTS is specifically designed to capture deep structural and texture similarities, which inherently covers much of the information provided by LPIPS (deep features) and ERQA (edge-based quality). Tables 15 and 16 show that adding these extra features yields only marginal gains in F1-score and PR-AUC. From a "lightweight" perspective, what specific, unique artifact characteristics do LPIPS and ERQA capture that are not already accounted for by DISTS? what was the specific motivation for including these redundant features, and do you consider the resulting computational cost to be a justifiable trade-off for such a marginal performance improvement?

**Limitations:**

1. Boundary Ambiguity: As noted by the authors, delineating exact artifact boundaries is inherently ambiguous for human annotators, which makes the binary masks used for training and evaluation only approximate

2. Ambiguity in Artifact Categorization: The proposed method merges all distortions into a single score, failing to distinguish between visually unpleasant artifacts (e.g., aliasing) and plausible high-frequency textures that simply deviate from the ground truth. This lack of specificity makes it unclear if the model is penalizing genuine quality loss or merely "plausible hallucinations," significantly limiting its diagnostic utility for SR refinement.

**Strengths And Weaknesses:**

Strengths:
   1. First SISR Artifact Dataset with Perceptual Prominence: The introduction of a specialized dataset containing 1,302 SISR artifact examples with crowdsourced prominence annotations is a notable contribution. By addressing the gap in existing binary-masked datasets, this work reveals that nearly half of previously identified artifacts are not actually prominent to human viewers.
   2. Effective Multi-Feature Fusion: The proposed prominence-modeling method successfully aggregates diverse features from existing metrics (DISTS, ssm_jup, and bd_jup) through a lightweight MLP regressor. This ensemble approach achieves better alignment with human perception than any individual metric, providing a more reliable spatial prominence heatmap.
   3. Practical Pseudo-GT Framework: The framework utilizes a lightweight SR model (RLFN) as a pseudo-ground truth to adapt full-reference metrics for real-world scenarios where original HR frames are unavailable. It provides a functional baseline for applying perceptual metrics in practical upscaling pipelines where no ground truth exists.

Weaknesses:
   1. Statistical Insignificance of Postprocessing: The proposed mask-postprocessing algorithm yields a improvement in mean artifact prominence (49.4% vs. 47.7%). Given that the paper admits a 95% confidence interval of approximately ±20% with 30 assessors, this 1.7%p gain is statistically negligible and fails to justify the necessity of the step.
   2. Questionable Utility of the Prominence Distribution: The authors define $\kappa=0.3$ as the threshold for notable artifacts that warrant detection. However, 47% of the collected samples fall below this threshold, meaning nearly half of the manually refined dataset consists of visually insignificant distortions. This raises a critical concern: if the primary goal is to detect prominent artifacts, does the distribution of the remaining 53% of valid samples offer any meaningful diversity or coverage beyond existing binary datasets (like DeSRA)? Without a detailed analysis of the sample density in the high-prominence range (e.g., 0.7–1.0), it remains unclear whether the proposed dataset provides a superior training signal or simply replicates existing benchmarks with an added layer of low-level noise.
   3. Dependency on Pseudo-GT Fidelity: The framework's robustness depends on the lightweight SR model's ability to avoid generating its own textures; when RLFN fails to reconstruct fine textures, it leads to false-positive artifact detections. This limitation highlights a vulnerability to domain shifts where the pseudo-GT model might underperform.
   4. Methodological Redundancy: The framework relies on overlapping off-the-shelf metrics (DISTS, LPIPS, ERQA) combined with a simple 3-layer MLP. This architecture exhibits significant conceptual redundancy, as DISTS already accounts for the deep structural features provided by LPIPS. Furthermore, core components like $ssm\_jup$ are direct adaptations of existing detectors (e.g., LDL), offering limited technical novelty beyond the simple fusion of redundant features.

---

> ### Author Rebuttal · Authors · 2026-03-31
>
> Thank you for your thorough review and in-depth observations! We’d like to address the weaknesses and questions.
>
> > Statistical Significance of Post-processing
>
> The primary motivation for post-processing is making sparse metric outputs amenable to subjective evaluation. For the DeSRA dataset this is less of a concern as their hand-made annotations are dense, but for our masks obtained from existing metrics it’s much more important.
>
> The raw DeSRA metric predictions are also somewhat sparse, so DeSRA actually does a similar morphologic postprocessing step to their metric outputs. The difference is that we do our postprocessing step consistently for all tested methods, and specifically for subjective evaluation.
>
> The near-identical mean prominence to no postprocessing on DeSRA indicates that our procedure doesn’t introduce any systematic bias.
>
> > Questionable Utility of the Prominence Distribution
> >
> > Without a detailed analysis of the sample density in the high-prominence range (e.g., 0.7–1.0), it remains unclear whether the proposed dataset provides a superior training signal or simply replicates existing benchmarks with an added layer of low-level noise.
>
> The sample density is 30% in 0.7-0.8, 34% in 0.8-0.9 and 36% in 0.9-1.0.
>
> We think that the artifact difference in the high-prominence range (0.7+) is perhaps less interesting compared to what the rest of the prominence data allows us to do with the dataset. For example, we can define an arbitrary detection goal, e.g. that we want to detect artifacts noticed by at least 15% of human observers, and train/validate our method accordingly (i.e. set κ=0.15). Or, we could omit the middle range (40%-60% prominence) from the training set entirely as noisy/unreliable data. Binary masks fundamentally do not allow this downstream flexibility.
>
> > Mitigating Dataset Bias: Section G and Section 3.3 acknowledge a bias toward low-prominence samples, with nearly half of the masks falling below 0.3 prominence. How did you ensure that the MLP regressor learned to identify truly disturbing, high-prominence artifacts rather than simply overfitting to low-level texture noises that are barely visible to humans?
>
> Our MLP learns to predict the prominence values so it’s not treating those non-prominent samples as artifacts. Prominence values close to 0 are essentially signalling an absence of artifacts in training, which is a valuable signal in itself, to avoid overfitting to the noise and detecting any SR input as artifact.
>
> Regarding the class imbalance, please see our Appendix I.4 where we try weighting the dataset samples to account for it. The result showed very similar performance to our final model.
>
> As a concrete proof that our model detects high-prominence artifacts, see our subjective experiment in Section 5.4 and Table 7.
>
> > Pseudo-GT Reliability in High-Frequency Regions
>
> There is a slight risk of misdetection which we discuss in Appendix E and Figure 13, but this failure case doesn’t occur frequently in practice from our testing. Please refer to our rebuttal to uhQZ where we show an additional SeeSR reference experiment.
>
> With higher-resolution input (e.g. the DeSRA dataset), this becomes even less of a concern as it becomes easier for RLFN to restore fine details. This is corroborated by Tables 2 and 3 showing more similar results between RLFN and MSE-SR on DeSRA.
>
> > Conceptual Overlap and Feature Redundancy
> >
> > From a "lightweight" perspective, what specific, unique artifact characteristics do LPIPS and ERQA capture that are not already accounted for by DISTS?
>
> We found that ERQA, unlike DISTS and LPIPS, is less prone to falsely flagging smooth textures such as concrete and asphalt as artifacts, meanwhile it can correctly flag some edges that the other two miss. LPIPS appears better than the other two at identifying false blurring and false deblurring by the SR model. So, we believe that all 3 features provide valuable signals and justify their added computational cost.
>
> > Ambiguity in Artifact Categorization: The proposed method merges all distortions into a single score, failing to distinguish between visually unpleasant artifacts (e.g., aliasing) and plausible high-frequency textures that simply deviate from the ground truth.
>
> This is a great observation: we don’t want to mark plausible hallucinations as prominent artifacts. We accounted for this concern in our subjective prominence annotation methodology. We show the participants a nearest-neighbor-upscaled image as the “Original”, rather than the ground truth. This is done specifically to avoid participants marking plausible, but different from original, hallucinated details as artifacts.
>
> Our pseudo-GT approach also helps since the metrics never see the original full-resolution GT and cannot make false inferences based on it.

---

> > ### Author Rebuttal · Reviewer_f3QX · 2026-04-02
> >
> > I appreciate the authors' detailed rebuttal. The explanations regarding the use of the dataset bias (low-prominence samples) as a negative signal to prevent overfitting, and the use of NN-upscaled images to avoid penalizing plausible hallucinations, are technically sound and successfully address my initial concerns in these specific areas.
> > **However, a few of my concerns remain only partially resolved, and I encourage the authors to address the following points in the final version. Assuming these revisions are thoughtfully incorporated, I am willing to maintain my recommendation of Weak Accept.**
> > The following points remain important for the final version:
> > 1. Calibration of Claims on Post-Processing: I understand that mask post-processing is a common morphological technique, and I agree with your point that it does not introduce systematic bias. However, it is overclaimed with a 1.7%p improvement that the mask post processing was helpful without further quantitative evidence (e.g., reduced evaluation time or increased inter-rater agreement). I suggest revising the text to neutrally state that the processing was applied for visual formatting without introducing systematic bias, rather than framing it as a direct improvement.
> > 2. Justification for Feature Redundancy (Computational Cost): The intuitive explanation for using LPIPS and ERQA alongside DISTS (e.g., reducing false positives on smooth textures) is helpful. However, to better justify the multi-feature architecture and the resulting computational overhead, it would strengthen the paper to provide a bit more supporting evidence. For example, this could be supported by:
> > •	A table comparing the exact computational cost (FLOPs/Parameters) and Inference Time (ms) of using DISTS alone versus the combined features.
> > •	Visual ablation examples (DISTS-only vs. All features) that explicitly demonstrate the claimed benefits on specific textures.
> > 3. Better Evidence of Diversity in the High-Prominence Range: While I understand the flexibility of a continuous score, the score alone may not fully demonstrate a meaningfully richer training signal. It would improve the paper to include a comparative analysis between the proposed dataset and the existing DeSRA dataset. Specifically, demonstrating how the prominence distribution of high-prominence artifacts differs, alongside visual examples showing new, diverse, or more complex distortions that might be missing in DeSRA, would help better justify the dataset’s unique value over existing binary-mask benchmarks.
> > 4. Main-text discussion of pseudo-GT limitations: I appreciate that the authors acknowledge the pseudo-GT failure cases in the appendix. However, the risk of penalizing high-quality SR models due to the pseudo-GT's limitations is a structural limitation of this framework. It would be beneficial for future users if the authors briefly discussed potential mitigation strategies in the main text (e.g., in a Future Work or Discussion section). For instance, mentioning how the framework might evolve or resolve this issue as more advanced reference models become available would provide a much more transparent and constructive outlook.
> > 5. Explicit Disclosure of the NN-Upscale Methodology: While using NN-upscaled images instead of the actual Ground Truth to prevent participants from penalizing 'plausible hallucinations' is a sound methodological choice, this important detail appears to be missing from Section 3.1 of the submitted manuscript. To ensure reproducibility and a correct understanding of the dataset's nature, I recommend that the authors explicitly detail this methodology in the main text, clearly defining what the "Original" image refers to in the annotation setup and explaining the purpose behind this choice.
> > Conclusion: The paper provides a meaningful contribution to the field of SR artifact detection. Addressing the points above will ensure greater transparency and methodological rigor, which will help fully justify the impact of this valuable work.

---

> > > ### Author Response · Authors · 2026-04-08
> > >
> > > Thank you for your acknowledgement. We will incorporate changes into the final version as per your suggestions:
> > >
> > > 1. We will revise the postprocessing text to state the primary goal was visual presentation, with an example if the page limit allows.
> > > 2. We will add a computational cost comparison table to the appendix along with further ablation examples.
> > > 3. We will add an appendix section with our training on DeSRA experiment from our response to Rf4H. In this experiment we train our method on the DeSRA dataset (with our prominence annotations) and find that its performance on less prominent artifacts degrades compared to training on our proposed dataset. This will demonstrate the utility of our proposed dataset for training a better artifact detector. We will also include visual examples of more complex distortions in our dataset compared to DeSRA.
> > > 4. We will include pseudo-GT failure mitigation strategies in the main text as you suggest.
> > > 5. We will detail our “Original” image selection methodology and justification in the main text.

---

### Official Review · Reviewer_kzLs · 2026-03-15

**Soundness:** 3
**Presentation:** 2
**Significance:** 2
**Originality:** 2
**Overall Recommendation:** 3
**Confidence:** 4

**Summary:**

This paper regards generative defects as "perceptual prominence-aware artifacts" for addressing the artifact issue in generative single-image super-resolution (SISR) techniques. It constructs a dedicated dataset with binary masks and prominence annotations, designs a lightweight detection model for prominent artifact detection, and conducts evaluations on its accuracy, generalization, and two downstream tasks. Experimental results demonstrate that the proposed method outperforms comparative approaches and can effectively detect artifacts generated by SISR.

**Compliance With Llm Reviewing Policy:**

Affirmed.

**Key Questions For Authors:**

Please see the Weakness.

**Limitations:**

yes

**Strengths And Weaknesses:**

**Strength**
1. The paper proposes a perceptual prominence-based modeling method for SR artifact detection, demonstrating that targeted utilization of prominent regions can effectively enhance the accuracy and generalization of detection models.
2. The paper constructs and opens up a small dedicated dataset, clearly elaborating on the annotation process and details to ensure verifiability and reproducibility.
3. The paper features a comprehensive and rigorous experimental design, fully validating the method's effectiveness through evaluations across multiple datasets, generalization analysis across different SISR methods, optimization of SISR models based on the proposed method, and ablation experiments.


**Weakness**
1. The contributions are incremental, mainly transferring the perspective of prominence detection to this task, and the design of the lightweight model is mostly a combination of modules.
2. The SR models evaluated in the experiments are somewhat dated; consequently, the performance of the proposed method on more recent, state-of-the-art (SOTA) SR architectures has not yet been established.
3. The dataset is derived by uniformly applying 4× bicubic downsampling to original images to obtain low-resolution ones, indicating that the degradation mode is solely interpolative downsampling. However, there are various degradation modes for SR, so the generalization ability of the proposed method to other degradation modes remains unclear.
4. Why only binary 0/1 labels are used to indicate prominence (presence or absence) during annotation instead of setting more grading levels? This is not quite consistent with the way the designed model outputs prominence probabilities.

---

> ### Author Rebuttal · Authors · 2026-03-31
>
> Thank you for your review! We’d like to address the weaknesses and questions.
>
> > The contributions are incremental, mainly transferring the perspective of prominence detection to this task, and the design of the lightweight model is mostly a combination of modules.
>
> The prominence formulation to the SR artifacts problem, and the first prominence-annotated SR artifact dataset, are novel and important contributions. As we show in Section 3.3, more than half of lab-annotated SR artifacts in the DeSRA dataset turned out to be unnoticeable to most viewers, meaning that training and evaluating methods on binary artifact masks risks producing biased models and results that skew towards unnoticeable artifacts, limiting their practical utility. This is further confirmed by our SR fine-tuning experiment in Section 5.5 and Table 6 where our method (that used prominence annotations for training) achieved better artifact reduction compared to DeSRA.
>
> > The SR models evaluated in the experiments are somewhat dated; consequently, the performance of the proposed method on more recent, state-of-the-art (SOTA) SR architectures has not yet been established.
>
> We appreciate this concern. However, we note that our chosen SR models are actively used as the primary baselines in the latest top-venue research. For example:
>
> - Dong, Linwei, et al. "TSD-SR: One-step diffusion with target score distillation for real-world image super-resolution." CVPR 2025: compares against SUPIR, StableSR, SeeSR, SinSR, ResShift.
> - Li, Jianze, et al. "Unleashing the Power of One-Step Diffusion based Image Super-Resolution via a Large-Scale Diffusion Discriminator." NIPS 2025: compares against SinSR, StableSR, SeeSR, ResShift, Real-ESRGAN, SwinIR.
> - Sun, Lingchen, et al. "Improving the stability and efficiency of diffusion models for content consistent super-resolution." IEEE TIP, December 2025: compares against StableSR, SeeSR, SinSR, SUPIR, SwinIR, Real-ESRGAN, ResShift.
>
> We believe this demonstrates that our SR method choice aligns with what the community currently considers state-of-the-art.
>
> > The dataset is derived by uniformly applying 4× bicubic downsampling to original images to obtain low-resolution ones, indicating that the degradation mode is solely interpolative downsampling. However, there are various degradation modes for SR, so the generalization ability of the proposed method to other degradation modes remains unclear.
>
> We have chosen 4x bicubic downsampling (with correct pre-filtering to avoid aliasing) as our degradation mode as this is one of the standard evaluation protocols used in SR research. Exploring the Real-ESRGAN degradation pipeline, as well as using the original photos as inputs (no synthetic degradation), would be valuable experimental settings for future work.
>
> Please also refer to Appendix H where we evaluated our method on the learning-based compression task (using the JPEG AI codec) where no synthetic degradation took place. Our method has proven to generalize well to this novel domain, considering it had not seen any learning-based compression artifacts in training.
>
> > Why only binary 0/1 labels are used to indicate prominence (presence or absence) during annotation instead of setting more grading levels? This is not quite consistent with the way the designed model outputs prominence probabilities.
>
> As we describe in Section 3.1, in our crowdsourced annotation, each image is ranked by 30 different participants, and the resulting prominence is the proportion of people who voted that an artifact is present. Effectively, this gives 31 grading levels to every artifact in our dataset. This setting is consistent with established pairwise comparison practices.
>
> Asking participants to rate artifacts on a more granular scale would introduce inter-annotator calibration issues, furthering the inter-annotator disagreement that we analyze in Appendix B.

---

> > ### Author Rebuttal · Reviewer_kzLs · 2026-04-04
> >
> > Thank you for the rebuttal. W2 and W4 are resolved. W1 is partially resolved — the dataset contribution is acknowledged. W3 is not resolved: the dataset covers only 4× bicubic downsampling, and I remain concerned about generalization to realistic mixed degradations (e.g., noise, blur, compression) commonly encountered in practical SR applications. Therefore, I maintain my original score.

---

> > > ### Author Response · Authors · 2026-04-08
> > >
> > > Thank you for your acknowledgement.
> > >
> > > Regarding W3: another point to consider is that adding more degradations, like noise, blur and compression, could lead to SR models producing more artifacts. This could happen when a given SR model hasn’t been trained for heavier degradations, or just as the result of the added uncertainty/noise on already-difficult input images.
> > >
> > > In fact, we had seen something similar happen when trying to use naive 4× bicubic downsampling instead of downsampling with correct prefiltering/antialiasing: while for a human observer such low-resolution inputs looked fine, several models produced much more severe artifacts on such inputs, presumably because naive bicubic downsampling was out-of-domain for those models. Notably, not all models exhibited artifacts, indicating that some models did train on naive downsampling.
> > >
> > > Our method did detect those artifacts successfully. However, while these artifacts are in scope for our method, we decided to focus on correct prefiltered downsampling as a harder and more realistic case. We think that if our method can find artifacts in the setting where the SR models behave best, then it will generalize well to real-world inputs in the downstream tasks.
> > >
> > > The same argument can be made about mixed degradations. While it’s certainly a valuable experimental setting, we believe that our chosen standard 4× bicubic downsampling with prefiltering, together with our generalization experiments, already serves well to demonstrate the practical utility of our method.
> > >
> > > Please also note that, apart from the already-mentioned JPEG AI in Appendix H, we demonstrate the generalization ability of our method across SR architecture families in Table 4 in Section 5.3. The difference between running the same SR on bicubic vs. mixed degradation input is likely less than running an SR from an architecture family unseen during training.

---

### Official Review · Reviewer_uhQZ · 2026-03-16

**Soundness:** 3
**Presentation:** 3
**Significance:** 2
**Originality:** 3
**Overall Recommendation:** 4
**Confidence:** 4

**Summary:**

This paper first introduces a dataset of 1302 SR artifact examples with crowdsourced prominence scores to address the problem of detecting artifacts in generative SISR. The prominence score aims to characterize artifacts by their perceptual impact rather than binary masks.

Building on this dataset, the paper also proposes a prominence-aware artifact detection approach that predicts spatial prominence heatmaps. The method extracts several distortion features between the SR output and a reference image.

Experiments show that the proposed approach achieves improved performance compared with existing artifact detection methods, generalizes across different SR architectures, and can be used to guide artifact suppression during SR model fine-tuning.

**Compliance With Llm Reviewing Policy:**

Affirmed.

**Key Questions For Authors:**

1. Could the authors elaborate on how sensitive the method is to the choice of this pseudo-reference model? Would the detection performance degrade if the pseudo-reference itself contains artifacts?

**Limitations:**

The paper would benefit from a clearer discussion of its limitations. In particular, it does not sufficiently acknowledge the simplifying assumption in the ground-truth annotation process, which may overlook fine-grained spatial variations.

**Strengths And Weaknesses:**

Strengths:
1. The paper addresses a real yet underexplored evaluation problem. Although only a few prior works have studied artifact detection in SR, this task is critical for making generative SR models practically usable. In addition, the main premise, that artifact localization should account for human noticeability rather than just binary presence, is sensible and well motivated.
2. Substantial experimental and annotation effort. The experiments include comparisons with multiple baselines, cross-model generalization tests, and analyses of the proposed metric and its downstream applications, providing strong empirical support for the proposed approach.

Weakness:
1. The authors claim to predict "spatial artifact prominence heatmaps", but the method for generating the ground truth lacks true spatial granularity. According to Section 3.1, it assumes that the prominence of an artifact is uniformly distributed across the entire annotated region. In practice, however, the artifact is localized and varies significantly pixel-by-pixel or patch-by-patch even within a single distorted region.
2. The artifact masks are obtained by running existing quality metrics. This raises a potential bias: the dataset may favor artifacts that these methods can already detect, while artifacts that they fail to capture may be underrepresented.

Suggestions For Rebuttal:
The proposed idea is promising and supported by empirical results on the SR task; however, the paper suffers from limitations in the design of the ground-truth annotations and potential bias in the dataset construction process. Addressing these concerns would significantly strengthen the paper, and I would be willing to reconsider my rating accordingly.

---

> ### Author Rebuttal · Authors · 2026-03-31
>
> Thank you for your review! We’d like to address the weaknesses and questions.
>
> > The authors claim to predict "spatial artifact prominence heatmaps", but the method for generating the ground truth lacks true spatial granularity. According to Section 3.1, it assumes that the prominence of an artifact is uniformly distributed across the entire annotated region. In practice, however, the artifact is localized and varies significantly pixel-by-pixel or patch-by-patch even within a single distorted region.
>
> We observed that regions in our dataset generally enclose a single artifact that is self-similar across the entire region with no significant variation (see examples on Figures 1, 2, 4, 11). The artifacts are generally constrained to a single object or texture. A single prominence value per region is therefore a valid assumption. Moreover, obtaining pixel-wise prominence annotations via crowdsourcing would be very costly (for example, by dividing the candidate region into patches, and asking the participants to assess every patch separately).
>
> Our region-level score is a good approximation that already reveals the inadequacy of the binary labels: in Section 3.3 we find that half of the artifacts in the lab-annotated DeSRA dataset are unnoticeable to most viewers.
>
> > The paper would benefit from a clearer discussion of its limitations. In particular, it does not sufficiently acknowledge the simplifying assumption in the ground-truth annotation process, which may overlook fine-grained spatial variations.
>
> We acknowledge this limitation in our Conclusion on lines 428-430. We will expand it in the revised paper as follows:
>
> > The artifact masks are obtained by running existing quality metrics. This raises a potential bias: the dataset may favor artifacts that these methods can already detect, while artifacts that they fail to capture may be underrepresented.
>
> We agree that obtaining masks using any single given quality metric would produce a biased dataset. For this reason, for our dataset we used 8 diverse metrics (SSIM, DISTS, LPIPS, bd_jup, ssm_jup, LDL, DeSRA, ours) — both classical and learning-based, targeting both image quality assessment and artifact detection — as well as manual annotation (constituting 14% of all masks). We believe this multi-source strategy substantially mitigates bias from any single metric. Moreover, any potential bias in the masks doesn’t diminish the main value of our work: the crowdsourced prominence annotations, which are novel ground truth.
>
> > Could the authors elaborate on how sensitive the method is to the choice of this pseudo-reference model? Would the detection performance degrade if the pseudo-reference itself contains artifacts?
>
> We explore this question in Appendix E. Figure 13 shows example false-detections when RLFN (our choice pseudo-reference) fails to restore sharp textures. That said, after running several metrics to detect artifacts on RLFN itself (similarly to the SR robustness experiment in Section 5.4), we haven’t found RLFN to produce many such artifacts or failure cases in practice.
>
> Further, please refer to tables 2, 3 in the main text, and 15-19, 21-24 in the appendix, which compare across pseudo-reference inputs: RLFN, SPAN, DeSRA’s MSE-SR, and  Original HR. While there’s a slight performance drop compared to Original HR, the detection remains robust. This is further corroborated by our method achieving strong results in all other experiments that use RLFN.
>
> At your suggestion, we’ve run several of the metrics, and our scoring, using SeeSR as the pseudo-reference (ranked 5th worst in Table 5 by artifacts produced). Here are the results for our full dataset:
>
> |               | Original HR  | Original HR | RLFN         | RLFN       | SeeSR        | SeeSR      |
> | ---           | ---          | ---         | ---          | ---        | ---          | ---        |
> | **Method**    | **F1-score** | **PR-AUC**  | **F1-score** | **PR-AUC** | **F1-score** | **PR-AUC** |
> | DISTS         | *0.0555*     | 0.0062      | **0.0706**   | *0.0082*   | **0.0646**   | 0.0063     |
> | bd\_jup       | 0.0043       | 0.0027      | 0.0074       | 0.0017     | 0.0025       | 0.0031     |
> | ssm\_jup      | 0.0251       | 0.0012      | 0.0180       | 0.0009     | 0.0135       | 0.0005     |
> | DeSRA         | 0.0405       | *0.0068*    | 0.0315       | **0.0120** | *0.0458*     | **0.0148** |
> | Ours (t=0.15) | 0.0355       | **0.0121**  | 0.0312       | 0.0075     | 0.0168       | *0.0064*   |
> | Ours (t=0.3)  | **0.0559**   | **0.0121**  | *0.0334*     | 0.0075     | 0.0295       | *0.0064*   |
>
> As expected, using a pseudo-reference itself prone to artifacts degrades the metric performance somewhat, likely owing to false positive detections.

---

### Decision · Program_Chairs · 2026-04-30

**Decision:**

Reject

**Comment:**

This paper proposes a method to detect SR artifacts with perceptual prominence, which is important for SR. It receives two weak rejects and two weak accepts. After reading the rebuttal, there are several major issues remaining. First of all, one of the major contributions is the prominence modeling for artifact detection. However, how the algorithm benefits from it is not clear. Even though additional experiments are added in the rebuttal, a more detailed analysis is needed. In addition, as pointed out by uhQZ, the dataset may favor artifacts that existing quality metrics can already detect. Although the authors claim that the multi-source strategy with manual annotation substantially mitigates bias, more analysis and experiments are needed to validate that. What is more, the proposed method has so many components with not sufficient ablation study, it is difficult to tell the benefit of each component. Generally, artifact detection is very useful for SR task and the authors provide a method. However, there are still many major issues in the current submission. As a result, it cannot be accepted in the current form.